# Lung SORT LNPs enable precise homology-directed repair mediated CRISPR/Cas genome correction in cystic fibrosis models

Tuo Wei [1,2,3,7], Yehui Sun [1,2,3,7], Qiang Cheng [1,2,3,7], Sumanta Chatterjee[1,2,3], Zachary Traylor[4,5], Lindsay T. Johnson [1,2,3], Melissa L. Coquelin[6], Jialu Wang[6], Michael J. Torres[6], Xizhen Lian[1,2,3], Xu Wang [1,2,3], Yufen Xiao [1,2,3], Craig A. Hodges[4,5] & Daniel J. Siegwart [1,2,3] ✉

Approximately 10% of Cystic Fibrosis (CF) patients, particularly those with CF transmembrane conductance regulator (*CFTR*) gene nonsense mutations, lack effective treatments. The potential of gene correction therapy through delivery of the CRISPR/Cas system to CF-relevant organs/cells is hindered by the lack of efficient genome editor delivery carriers. Herein, we report improved Lung Selective Organ Targeting Lipid Nanoparticles (SORT LNPs) for efficient delivery of Cas9 mRNA, sgRNA, and donor ssDNA templates, enabling precise homology-directed repair-mediated gene correction in CF models. Optimized Lung SORT LNPs deliver mRNA to lung basal cells in Ai9 reporter mice. SORT LNP treatment successfully corrected the *CFTR* mutations in homozygous G542X mice and in patient-derived human bronchial epithelial cells with homozygous F508del mutations, leading to the restoration of CFTR protein expression and chloride transport function. This proof-of-concept study will contribute to accelerating the clinical development of mRNA LNPs for CF treatment through CRISPR/Cas gene correction.

Although effective therapies have been developed for patients with gating, conduction defects, or two copies of the F508del mutation in the Cystic Fibrosis Transmembrane Conductance Regulator (*CFTR*) gene, most patients with two copies of nonsense mutations remain untreatable with current approaches[1-7]. Looking ahead to potentially curative therapies, CRISPR/Cas is a revolutionary genome editing technology[8-11] that if successfully developed to correct *CFTR* mutations could be a transformative advance resulting in long-lasting therapies for all Cystic Fibrosis (CF) patients, including those with nonsense mutations.

A key bottleneck is the lack of delivery strategies required to enable targeted editing in specific cells, especially cells in lungs[7,12,13]. To date, successful in vivo editing has been mediated mainly by viral vectors, which present challenges for clinical translation due to potential immunogenicity, concerns about dangerous integration events, and inability to re-dose[6,13]. Intravenous non-viral lipid nanoparticle (LNP) delivery offers advantages in those respects, but advances to date have been largely limited to targets in the liver[7,14-18].

We recently overcame this delivery challenge through the development of a class of non-viral nanoparticles for tissue-specific genome editing[19-21]. Selective ORgan Targeting (SORT) LNPs enable intravenous delivery of nucleic acids and proteins to the lungs, liver, and spleen, plus local cell-specific delivery to the muscle and brain. Importantly, we showed that Lung SORT LNPs enable genome editing

[1]Department of Biomedical Engineering, Program in Genetic Drug Engineering, The University of Texas Southwestern Medical Center, Dallas, TX, USA. [2]Department of Biochemistry, The University of Texas Southwestern Medical Center, Dallas, TX, USA. [3]Simmons Comprehensive Cancer Center, The University of Texas Southwestern Medical Center, Dallas, TX, USA. [4]Department of Genetics and Genome Sciences, Case Western Reserve University School of Medicine, Cleveland, OH, USA. [5]Department of Pediatrics, Case Western Reserve University School of Medicine, Cleveland, OH, USA. [6]ReCode Therapeutics, Menlo Park, CA, USA. [7]These authors contributed equally: Tuo Wei, Yehui Sun, Qiang Cheng. ✉e-mail: Daniel.Siegwart@UTSouthwestern.edu

in cells across the lungs, from the endothelium to the epithelium in reporter mice[19], which encouraged us to explore genome editing to treat cystic fibrosis in vivo following direct, intravenous (IV) administration. Separate from lung targeting, we recently demonstrated the ability to deliver multiple CRISPR/Cas components (Cas9 mRNA, sgRNA, and ssDNA template) for efficient HDR-mediated gene correction in vitro and in vivo to tumor xenografts[22,23], which further motivated the current study investigating lung-targeting SORT LNPs for HDR-based CF therapy.

To achieve this goal, we first examined a series of permanently cationic Lung SORT lipids, optimized the molar ratio of lipids components within LNPs, and ultimately identified improved LNP formulations for precise HDR genome editing ex vivo (DOTAP10 LNPs) and in vivo (DOTAP40 LNPs). The HDR correction efficiency in BFP/GFP switchable HEK293 cells with a Y66H mutation achieved as high as 50% and turned on bright GFP fluorescence in treated cells. Strikingly, optimized lung targeting LNPs (DOTAP40) could transfect ~60% of the lung basal cell population after systemic IV administration, which is very promising for CF therapy as sufficient *CFTR* gene correction in lung basal cells could contribute to long-term functional restoration of the airway epithelium. Optimized SORT LNPs successfully corrected the G542X *CFTR* mutation in mouse lungs of G542X CF mice and restored CFTR function in intestinal organoids derived from these animals. Additionally, SORT LNPs efficiently corrected the F508del mutation in patient-derived human bronchial epithelial (HBE) cells and restored the expression of CFTR protein and relative CFTR function after propagation. Overall, the strategy developed in this study is anticipated to accelerate the clinical development of mRNA and CRISPR/Cas approaches to treat CF by providing delivery tools and fundamental knowledge to advance genome correction strategies. Continued development of effective SORT LNPs to deliver CRISPR/Cas to CF-relevant organs and cell types in vivo may eventually enable a new therapy for CF.

## Results

### Optimization of SORT LNPs enabled enhanced mRNA delivery to mouse lungs with low toxicity

Development of a CRISPR/Cas HDR approach requires delivery of multiple components, including different nucleic acid types (Cas9 mRNA, sgRNA, and ssDNA) (Fig. 1). To improve delivery efficacy, we first reexamined the molar ratios of lipids used in LNPs to optimize them for higher lung-specific activity[19]. We determined that increasing the proportion of the ionizable amino lipid (5A2-SC8) in our previously

identified mDLNP formulation[24] (termed mDLNP-1) significantly enhanced mRNA delivery in vivo. Using the SORT lipid mixing strategy, we optimized lipid ratios to identify second-generation mDLNPs (termed mDLNP-2) with a molar ratio of 5A2-SC8/DOPE/Chol/PEG-DMG fixed at 36/20/40/4 (Fig. 2a). LNP formulations were examined for IV delivery of Luciferase (Luc) encoding mRNA in C57BL/6 mice. Luc activity was quantified 6 h after injection using in vivo luminescence imaging. mDLNP-2 enabled higher luminescence activity compared to mDLNP-1, indicating higher mRNA delivery efficiency (Figs. 2b, c).

Since SORT molecules for lung targeting fall into a generalizable class, we next screened a series of permanently cationic lipids with different chemical structures, including DOTAP, DDAB, DOTMA, EPC, and MVL5 (Supplementary Fig. 1). These cationic lipids were incorporated into mDLNP-2 with different percentages (fraction of total lipids). By delivering Luc mRNA, we compared the lung-targeting efficacy of these LNP carriers in vivo. The incorporation of DOTAP and DDAB in mDLNP-2 exhibited higher luminescence activity than others (Figs. 2d, e). To evaluate the lung targeting specificity, the relative luciferase expression of Lung/Liver and Lung/Spleen of different LNP carriers was calculated based on in vivo imaging data (Fig. 2f). All identified Lung SORT LNPs targeted the lungs effectively and hold promise for further development and optimization. Following an analysis of both lung delivery efficacy and specificity, DOTAP40 LNPs and DDAB30 LNPs were selected for further consideration.

During the evaluation process, we found that DDAB LNPs exhibited some toxicity after treatment. To further confirm this, in vivo toxicity of DOTAP40 and DDAB30 LNPs treatments were evaluated by measuring serum biomarkers relating to liver function (ALT and AST) and kidney function (BUN and CREA). Higher toxicity in the liver was observed in the DDAB30 LNP treatment group after 24 h (Fig. 2g, Supplementary Fig. 2). We also monitored the body weight changes and noted increased weight loss in mice treated with DDAB30 LNPs, suggesting higher acute toxicity (Fig. 2h). At the end of the study, mice were sacrificed, and organs (liver, spleen, and lung) were excised and weighed. The liver/body weight ratio and spleen/body weight ratio were increased with DDAB30 LNP treatment, suggesting higher toxicity (Fig. 2i). Immunohistochemistry (H&E staining) results showed that obvious tissue damages were detected in Lipopolysaccharide (LPS)-treated mouse organs and DDAB30 LNP-treated mouse organs (such as heart and liver), but not from DOTAP40 LNP treated mouse organs (Supplementary Fig. 3). To investigate the potential decrease in expression efficiency due to LNP immunogenicity after repeated

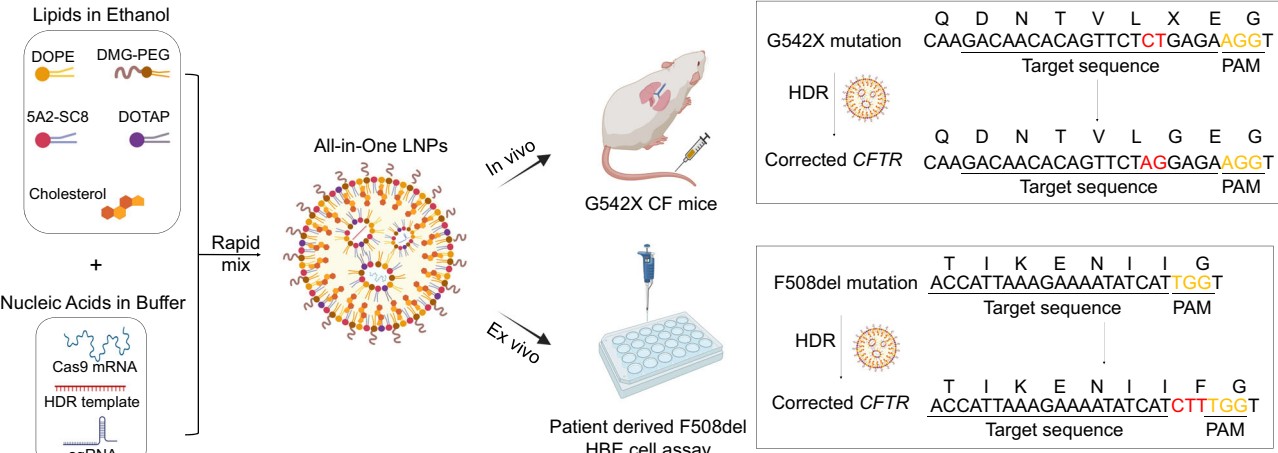

**Fig. 1 | Schematic illustration of SORT LNP-mediated gene correction therapy in CF mouse models.** SORT LNPs encapsulating three nucleic acid cargos (Cas9 mRNA, sgRNA, and ssDNA HDR template) effectively corrected *CFTR* gene mutations in a G542X CF mouse model (in vivo) and in a patient-derived F508del HBE cell model (ex vivo).

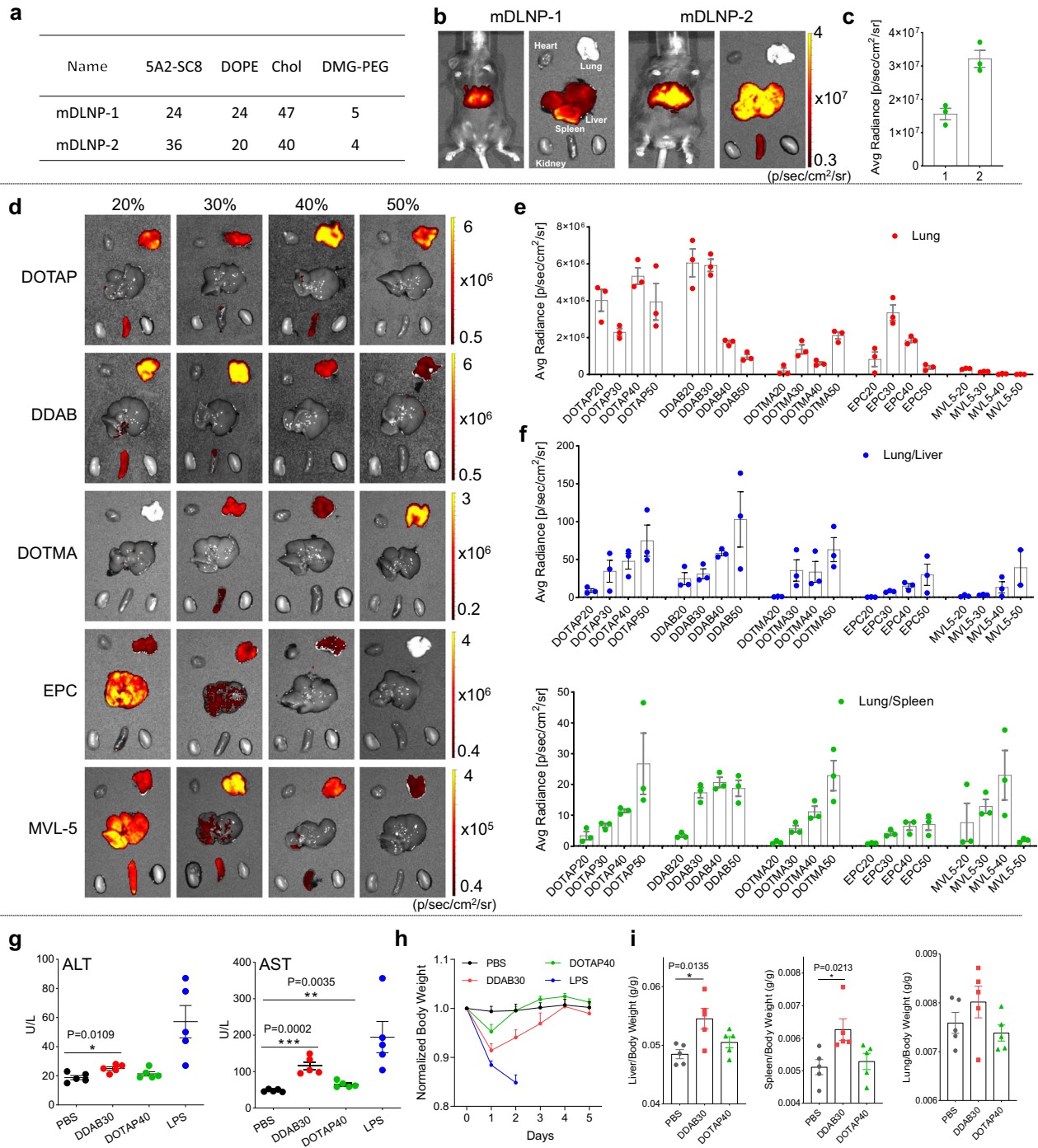

**Fig. 2 | Optimized DOTAP LNP formulation achieved enhanced mRNA delivery in mouse lungs with low toxicity. a** mDLNP-2 was optimized by adjusting the internal molar ratio among 5A2-SC8/DOPE/Chol/PEG-DMG to 36/20/40/4. **b** Higher luciferase expression in mouse liver was observed using mDLNP-2 compared to mDLNP-1, measured using IVIS (0.1 mg kg⁻¹ Luc mRNA, *i.v.*, 6 h). **c** Quantification data further confirmed enhanced mRNA delivery efficiency in mouse liver using mDLNP-2. Data are shown as mean ± s.e.m. (*n* = 3 biologically independent animals). **d** Permanently cationic lipid structures were screened for lung-targeting LNPs using IVIS. A series of 5A2-SC8 LNPs with 20% to 50% different permanently cationic lipids (fraction of total lipids) were prepared and administered into mice for testing (0.1 mg kg⁻¹ Luc mRNA, i.v., 6 h). **e** Quantification data suggested that DOTAP LNPs and DDAB LNPs showed higher luciferase expression in mouse lungs. **f** The lung-targeting specificity was evaluated by calculating the relative luciferase expression

of the Lung/Liver and Lung/Spleen. DOTAP40 and DDAB30 LNPs were selected by comparing lung delivery efficacy and specificity. Data are shown as mean ± s.e.m. (*n* = 3 biologically independent animals). **g** In vivo toxicity assessments revealed that DOTAP40 LNPs are less toxic compared to DDAB30 LNPs (1.5 mg kg⁻¹ mCherry mRNA, i.v., 40/1(total lipid/total RNA)), as indicated by liver function (ALT and AST) after 24 h of treatment (**g**), body weight changes after the treatment (**h**) and organ-to-body weight ratio (**i**). PBS treatment (i.v.) was used as a negative control, and lipopolysaccharide (LPS, i.p., 5 mg kg⁻¹) treatment was used as a positive control. LPS group mice were sacrificed at day 2, other groups were sacrificed at day 5. Data of **g**, **h**, **i** are shown as mean ± s.e.m. (*n* = 5 biologically independent animals). Source data are provided as a Source Data file. A two-tailed unpaired t-test was used to determine the significance of the comparisons of data (*P < 0.05; **P< 0.01; ***P< 0.001).

administrations, we administered DOTAP40 LNPs to C57BL/6 mice three times (once every week). The results demonstrated that the expression efficiency remained consistently high even after multiple dosing (Supplementary Fig. 4). In addition, DOTAP40 LNPs also allowed a moderate level of mRNA expression to other airway compartments (trachea and nasal epithelium) (Supplementary Fig. 5) Based on these findings, we selected DOTAP40 SORT LNPs (Supplementary Table 1) for continued study.

Because there are no animal models on the development path for CF drugs, most medicines are evaluated for efficacy in primary human bronchial epithelial (HBE) cells coupled with animal testing of tolerability and cellular tropism. Recognizing that delivery barriers in vivo are more extensive than delivery barriers in vitro, we envisioned using related DOTAP containing SORT LNPs to validate both in vivo and in vitro delivery capabilities. Our previous studies in this area demonstrated that optimal DOTAP LNPs for in vivo lung targeting were not ideal for in vitro cell studies or intramuscular injection[19,20]. To optimize DOTAP based SORT LNPs for in vitro utility, we prepared a series of mDLNP-2 SORT LNPs with 5%–30% DOTAP and examined Luc mRNA delivery to HeLa cells. The result showed that DOTAP10 was the most efficacious, producing the highest luminescence intensity after 24 h treatment (Supplementary Fig. 6). With this information in hand, we were able to proceed with in vitro, ex vivo, and in vivo studies.

## DOTAP10 LNPs enabled efficient gene correction in BFP/GFP switchable HEK293 cells with Y66H mutation

LNPs have been previously reported to efficiently edit cells by co-delivery of Cas9 mRNA and sgRNA in vivo, leading to gene knock-out by forming insertions and deletions (indels) following non-homologous end joining (NHEJ) repair pathway at the double-strand break (DSB) site[19,25,26]. However, this approach is not useful for genome correction. For CRISPR/Cas, precise gene correction at the mutated site by HDR is required to restore function. To evaluate the ability of DOTAP LNPs to deliver Cas9 mRNA, sgRNA, and ssDNA HDR templates for precise gene correction, we chose BFP/GFP switchable HEK293 cells as a reporter cell model. These cells expressing a GFP sequence have a single amino acid mutation (Y66H) in their GFP sequencing (TAC mutated to CAT), which alters the fluorescence from green (GFP) to blue (BFP)[27]. Once the mutation is corrected, the GFP function will be restored and the fluorescence will return to green (Fig. 3a).

Because the internal ratio among multiple CRISPR/Cas9 nucleic acid components is critical for HDR correction, we prepared a series of DOTAP10 LNPs with different weight ratios among Cas9 mRNA, sgRNA, and HDR template (corresponding molar ratios were listed in Supplementary Fig. 7). First, we fixed the weight ratio of Cas9 mRNA:sgRNA at 1:1, and adjusted the ratio of HDR template added. Sequencing results suggested that Cas9 mRNA:sgRNA:HDR at 1:1:6 had the highest HDR correction efficiency (Fig. 3b, Supplementary Fig. 7a, c). Next, we fixed the weight ratio of sgRNA:HDR at 1:6 and adjusted the ratio of Cas9 mRNA added. The second round of screening showed that the optimal weight ratio among Cas9 mRNA:sgRNA:HDR was 0.5:1:6, and the HDR correction efficiency reached as high as 50% (Figs. 3c, d, Supplementary Fig. 7b, c). Using the optimal internal ratio, DOTAP10 LNPs encapsulating Cas9 mRNA, sgRNA, and HDR templates (termed as DOTAP10-HDR) successfully corrected the mutated GFP gene and switched on bright GFP fluorescence in HEK293 cells, while little green signal was detected with treatment of DOTAP 10 LNPs encapsulating Cas9 mRNA and sgRNA (named DOTAP10-NHEJ) (Fig. 3e). Recognizing that inclusion of HDR templates may affect SORT LNP formation, we examined whether DOTAP40 LNPs could induce indels in different lung cell types of lox-stop-lox tdTomato (tdTOM) mice[19] following IV administration of Cas9 mRNA and sgRNA targeting the lox-stop-lox site (sgTOM1) with (DOTAP40-HDR) or without sacrificial ssDNA donor template (DOTAP40-NHEJ) (Fig. 3f). We observed obvious tdTomato signals both in bulk lung and different

lung cells type (including lung epithelial cells, lung endothelial cells, and lung immune cells) (Figs. 3g, h, Supplementary Fig. 8), with significantly higher editing levels in CFTR-associated lung epithelial cell populations than in the bulk lung. Moreover, we determined that mice receiving mRNA:sgRNA (Fig. 3g) or mRNA:sgRNA:ssDNA (Fig. 3h) had similar levels of editing in bulk lung or lung epithelial cells, suggesting that triple-loaded SORT LNPs retain editing capacity in vivo.

## Lung SORT LNPs could reach basal cells in mouse lungs

Encouraged by high HDR correction efficiency in HEK293 cells, we moved forward to treat CF mouse model harboring the G542X mutation in CFTR[28]. To maximize the dose of LNPs for in vivo study, we investigated whether decreasing the weight ratio between total lipid/total nucleic acid (NA) of DOTAP40 LNPs from 40:1 to 20:1 maintains efficacy, thereby improving tolerability of DOTAP containing LNPs. Comparable delivery efficiency in mouse lungs was observed after decreasing the weight ratio from 40:1 to 20:1 (Fig. 4a, Supplementary Fig. 9), indicating that less lipids could be used. Moreover, functional luciferase mRNA delivery efficiency was maintained predominately in lung after repeat dosing of DOTAP40-Luc up to three times in C57BL/6 mice and minimal off-organ editing mediated by co-delivery of Cas9 mRNA and sgTOM1 (DOTAP40-NHEJ) was observed in Ai14 mice (Supplementary Fig. 10). Therefore, we fixed the weight ratio of total lipid/total NA at 20:1 for DOTAP40 LNPs in the future in vivo studies.

Airway basal cells can differentiate into different cell types of airway epithelium. Correcting sufficient gene mutations in basal cells may contribute to long-term functional restoration of the airway epithelium[29–31]. Given this, the development of a delivery system that could deliver CRISPR/Cas9 system to basal cells would be a critical and meaningful advance for CF therapy. Previously, we reported that a single administration of Lung SORT LNPs (5A2-SC8 / DOTAP / DOPE / Cholesterol / PEG-DMG = 11.9/50/11.9/23.8/2.4 (mol/mol)) transfected ~60% of all endothelial cells and ~40% of all epithelial cells following a single IV injection of 0.3 mg kg$^{-1}$ Cre recombinase (Cre) mRNA to lox-stop-lox tdTomato mice[19]. Given that Lung SORT LNPs access epithelial cells from the blood side, we hypothesized that Lung SORT LNPs may also reach basal cells. To determine this, we quantified the transfection efficiency of optimized DOTAP40 LNPs encapsulating Cre mRNA (DOTAP40-Cre) to NGFR$^{+}$ basal cells using flow cytometry of cells extracted from edited mouse lungs following IV injection in tdTOM reporter mice (two injections, 2 mg kg$^{-1}$ Cre mRNA, 20/1 (total lipid/ total RNA)). Excitingly, DOTAP40-Cre transfected ~60% of NGFR$^{+}$ basal cell populations after treatment (Fig. 4b, Supplementary Fig. 11) This result is very promising and encouraged us further to evaluate the possibility of our delivery system to correct CFTR mutations in the lungs of a G542X CF mouse model.

## DOTAP LNPs successfully corrected G542X mutation in CF mouse lungs and restored CFTR function in an intestinal organoid model carrying G542X mutation

We next evaluated DOTAP40 LNPs for treatment of the CF mouse model with homozygous G542X nonsense mutation. DOTAP40 LNPs encapsulating Cas9 mRNA, sgRNA, and HDR templates targeting G542X mutation (DOTAP40-HDR) (total lipid/total NA = 20:1) were administered IV by tail vein into CF mice every week for a total of three times. One week after the last injection, whole mouse lungs were collected and homogenized for genome DNA extraction and next-generation sequencing (NGS) (Fig. 4c). CRISPResso2[32] analysis result based on deep sequencing showed that DOTAP40-HDR treatment successfully corrected G542X mutation in mouse lungs. Up to 2.34% of CFTR gene extracted from whole lung tissue was corrected (Fig. 4d). No off-target editing was detected by NGS and CRISPResso2 analysis (Supplementary Fig. 12) Given that we evaluated the gene correction level of the entire lung, it is possible that editing rates may have been

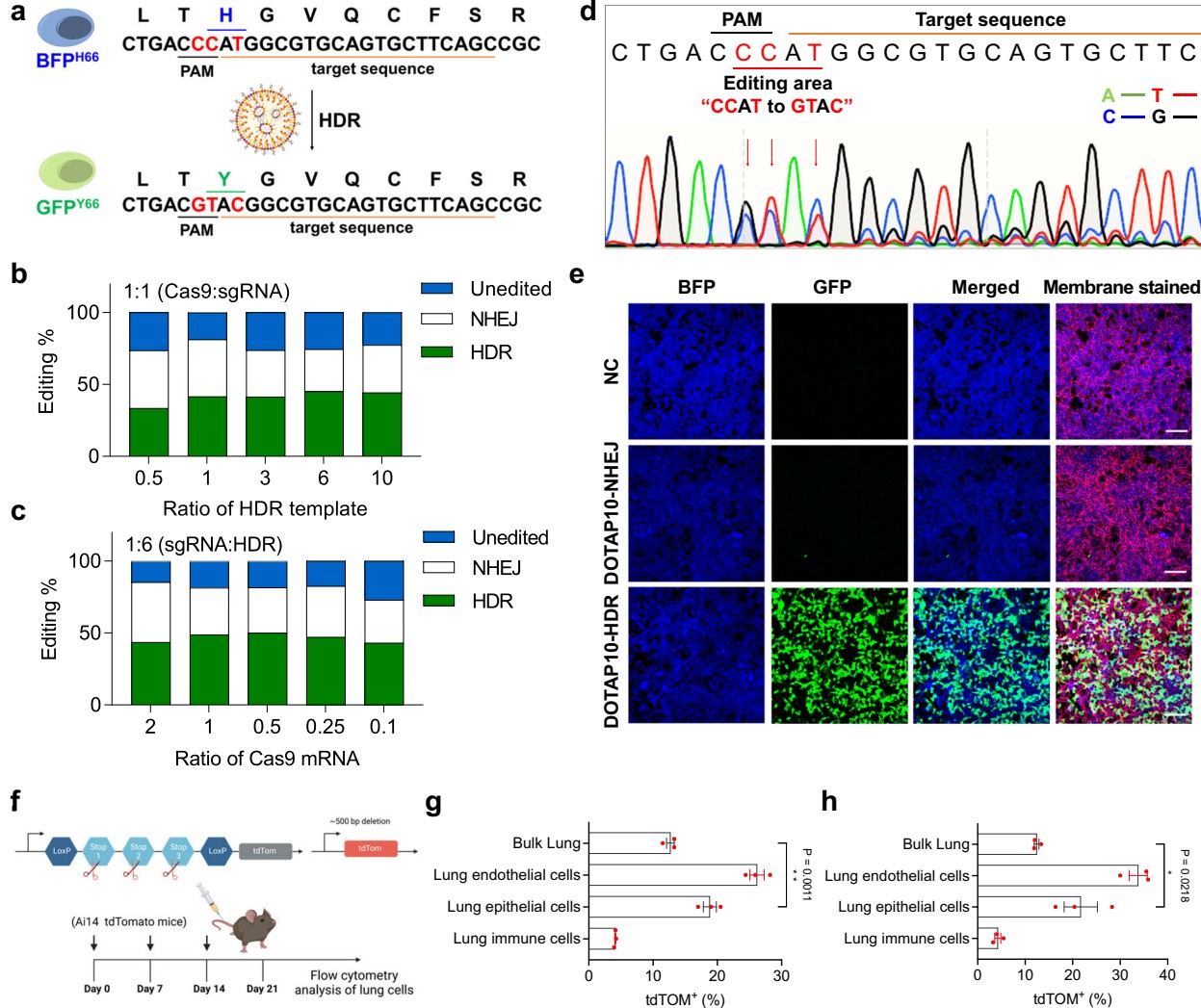

**Fig. 3 | DOTAP10 LNPs, encapsulating Cas9 mRNA, sgRNA, and ssDNA HDR template, successfully induced HDR in HEK293 cells with Y66H GFP mutation. a** The HEK293 cells have a Y66H mutation in the GFP sequence, which alters the fluorescence from GFP to BFP. Once corrected, the fluorescence will turn back to GFP. The gene editing efficiency was analyzed using TIDER analysis with DNA sequencing data obtained after testing with a series of DOTAP10 LNPs containing Cas9 mRNA, sgRNA, and donor ssDNA template with Cas9:sgRNA weight ratio fixed at 1:1 (**b**), then with sgRNA:HDR weight ratio fixed at 1:6 (total NA at 0.8 ng μL⁻¹) (**c**). The optimal weight ratio among Cas9 mRNA, sgRNA and HDR was 0.5/1/6 (named as DOTAP10-HDR). Data of **b**, **c** are shown as mean±s.e.m. (*n* = 3 biologically independent samples). **d**, DNA sequencing result after treatment of DOTAP10-HDR demonstrated that the editing area was successfully corrected from "CCAT" to "GTAC". **e** DOTAP10-HDR efficiently corrected Y66H mutation in HEK293 cells and turned on bright GFP fluorescence. NC group (PBS) and DOTAP10 LNPs

encapsulating Cas9 mRNA and sgRNA (named as DOTAP10-NHEJ) group were used as controls. The total NA was fixed at 0.4 ng μL⁻¹. Scale bar: 100 μm. The data was repeated three times independently with similar results. **f** Ai14 mice were used to evaluate Lung SORT LNP-mediated CRISPR/Cas gene editing in vivo. tdTOM mice were treated once a week with (**g**) DOTAP40-NHEJ (Cas9 mRNA:sgTOM1 = 2:1) or (**h**) DOTAP40-HDR (Cas9 mRNA: sgTOM1: HDR template = 2:1:3) formulations with a total NA at 1 mg kg⁻¹ IV. One week following the third injection, mice were sacrificed, and the lungs were collected for flow cytometry analysis to determine the proportion of tdTOM⁺ cells in different cell types. There is a significant difference in tdTom⁺ between bulk lung and lung epithelium cell populations with a *P* value equal to 0.0218 (DOTAP40-HDR). Data are shown as mean ± s.e.m. (*n* = 3 biologically independent animals). A two-tailed unpaired *t*-test was used to determine the significance of the comparisons of data (*$P$ < 0.05; **$P$ < 0.01; ***$P$ < 0.001; ****$P$ < 0.0001). Source data of **b**, **c**, **g**, **h** are provided in Source Data file.

diluted due to the presence of other cell types unrelated to CF. Considering the significantly higher editing efficiency of DOTAP40-HDR LNPs on lung epithelial cells compared to the bulk lung in tdTom mice (Fig. 3h), we anticipate that the gene correction at the G542X mutation site in CF relevant cells, such as lung epithelial cells, may be higher. This result is promising, as there are few dividing cell populations that can be corrected through HDR-mediated strategy. This requirement of CRISPR/Cas9 HDR is likely a limiting factor here, which may be overcome in future studies using next-generation gene editors that do not require double-strand breaks. The cloning result further confirmed that successful gene correction at G542X mutation site was observed with DOTAP40-HDR treatment (Fig. 4e).

Although the G542X mouse model contains G542X mutation in their genomic DNA, the lungs of these mice do not show CF-related pathological features[28]. Therefore, this model cannot be used to evaluate in vivo restoration of CFTR function. To overcome this limitation, we utilized intestinal organoids[28,33] derived from G542X mouse intestine stem cells as an ex vivo model to evaluate CFTR function recovery after DOTAP10-HDR treatment. The CFTR activator, forskolin, can stimulate intracellular pathways and phosphorylate CFTR, finally opening the CFTR channel and leading to ion/water uptake and organoid swelling. Once the mutated *CFTR* gene correction event occurs with organoids, organoid swelling should be observed using the forskolin-induced swelling (FIS) assay, while uncorrected

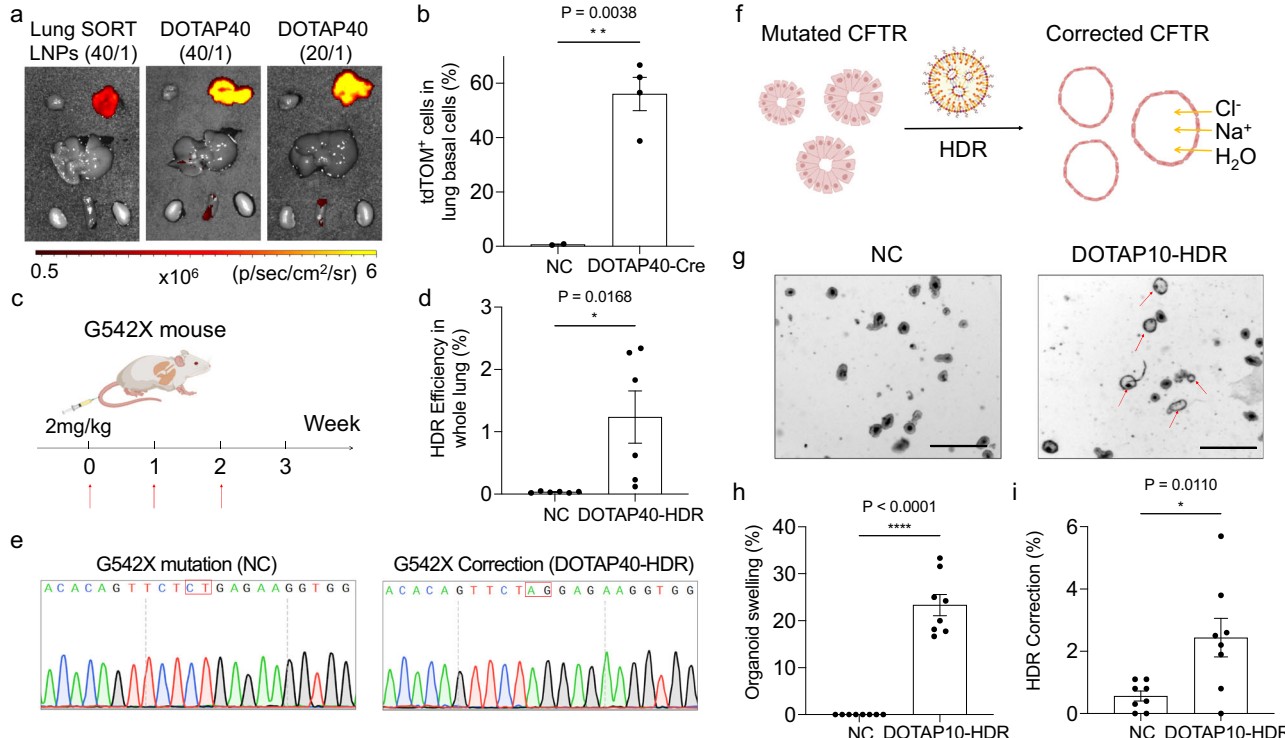

**Fig. 4 | DOTAP LNPs successfully corrected *CFTR* gene mutation in G542X-CF mouse model and restored CFTR function in a mouse intestinal organoid cell model with the G542X mutation. a** Optimized DOTAP40 LNPs showed higher luciferase expression in mouse lungs, compared to previously reported Lung SORT LNPs, and mRNA delivery efficiency remained comparable when the weight ratio of total lipid/total NA was reduced from 40/1 to 20/1 (0.1 mg kg⁻¹ Luc mRNA, $n = 3$ biologically independent animals). **b** DOTAP40 LNPs efficiently delivered Cre mRNA (2 mg kg⁻¹ Cre mRNA, 20/1) to basal cells of tdTOM mouse lung, activating tdTOM fluorescence. Two days after the second injection, the tdTOM positive cell in the mouse lung basal cell (NGFR +) population was analyzed using flow cytometry ($n = 4$ biologically independent animals in DOTAP40-Cre group). NP only group was used as negative control (NC). **c** DOTAP40-HDR LNPs encapsulating Cas9 mRNA, sgRNA, and HDR template were I.V. injected into CF mouse containing G542X mutation over three weeks (2 mg kg⁻¹ total NA, 20:1). **d** DOTAP40-HDR treatment successfully corrected G542X mutation in mouse lungs analyzed by NGS

deep sequencing and CRISPResso2 analysis. NP only group was used as negative control (NC). Data are shown as mean±s.e.m. ($n = 6$ biologically independent animals). **e** Representative sequencing result showed the successful HDR correction event at G542X mutation site was observed in mouse lungs after DOTAP40-HDR, by subsequent cloning and DNA sequencing. **f** Schematic illustration shows the mechanism of ex vivo intestinal organoid based forskolin-induced swelling (FIS) assay. **g** Intestinal organoid swelling was observed after treatment of DOTAP10-HDR, suggesting successful CFTR function restoration. No organoid swelling was detected after NC (NP only) treatment. Scale bar: 500 μm. The data was repeated three times independently with similar results. More than 20% of organoids swelled (**h**) and successful HDR correction by Sanger sequencing (**i**) were detected after treated with DOTAP10-HDR. Data are shown as mean ± s.e.m. ($n = 8$ biologically independent samples). Two-tailed unpaired t-tests were used to determine the significance of the comparisons of data (*$P < 0.05$; **$P < 0.01$; ***$P < 0.001$). Source data are provided as a Source Data File.

organoids (negative control, NC) will remain at the baseline volume (Fig. 4f)[34,35]. Although the isolated intestinal organoid delivery may not directly demonstrate in vivo lung cell delivery, the organoid swelling assay can serve as a valuable model to evaluate whether DOTAP LNPs can effectively facilitate gene correction and restoration of CFTR protein function. Quantitative results suggested successful CFTR function recovery with over 20% of organoids swelled after DOTAP10-HDR treatment (Figs. 4g, h). Subsequent TIDER analysis based on DNA sequencing further confirmed that there was *CFTR* gene correction event occurred (Fig. 4i). To note, the intestinal organoid swelling assay is a sensitive assay that can potentially magnify the extent of CFTR functional restoration. A previous study reported that moderate correction of *CFTR* mutation may lead to significant organoid swelling[36] which is consistent with our results (Figs. 4h, i). This result also highlights the capacity of DOTAP-LNPs for multi-cargo delivery to multiple cell types and utility in diverse cellular contexts.

**DOTAP LNPs corrected *CFTR* mutation and restored CFTR function in patient-derived human bronchial epithelial (HBE) cells with F508del *CFTR* mutation**

The F508del mutation is the most common CF disease allele, which accounts for 70% of all CF mutations[1]. Even though there are CFTR

modulators on the market that can help relieve disease related symptoms, there is still no way to cure CF disease permanently. Primary human bronchial epithelial (HBE) cells from CF patients with F508del mutation (F508del/F508del CF HBEs) are widely recognized as the gold standard assay for preclinical CF studies most predictive of clinical benefit[37,38]. Because F508del HBEs are more commonly available than G542X HBEs, we had to focus on correcting the F508del mutation in the HBE assay. Therefore, we next explored whether our LNP formulations could deliver RNA cargos to homozygous F508del/F508del HBEs (Fig. 5a). DOTAP10 LNPs encapsulating mCherry mRNA (DOTAP10-mCherry, 0.5 μg mL⁻¹) were transfected to F508del/F508del CF HBEs and their cellular uptake was evaluated using fluorescence microscopy. Excitingly, we observed bright mCherry fluorescence after transfection, compared to NC group (Fig. 5b) across the entire field of view. This demonstrated that our LNP formulation could effectively encapsulate and deliver RNA cargoes to HBE cells, providing potential for genome editing of CF mutations.

Then we explored the possibility of DOTAP10 LNPs to correct F508del mutation in HBE cells. Patient-derived undifferentiated HBE cells with homozygous F508del mutation (passage 1, P1) were treated with DOTAP10 LNPs encapsulating Cas9 mRNA, sgRNA, and HDR template (Supplementary Fig. 13) targeting the F508del mutation

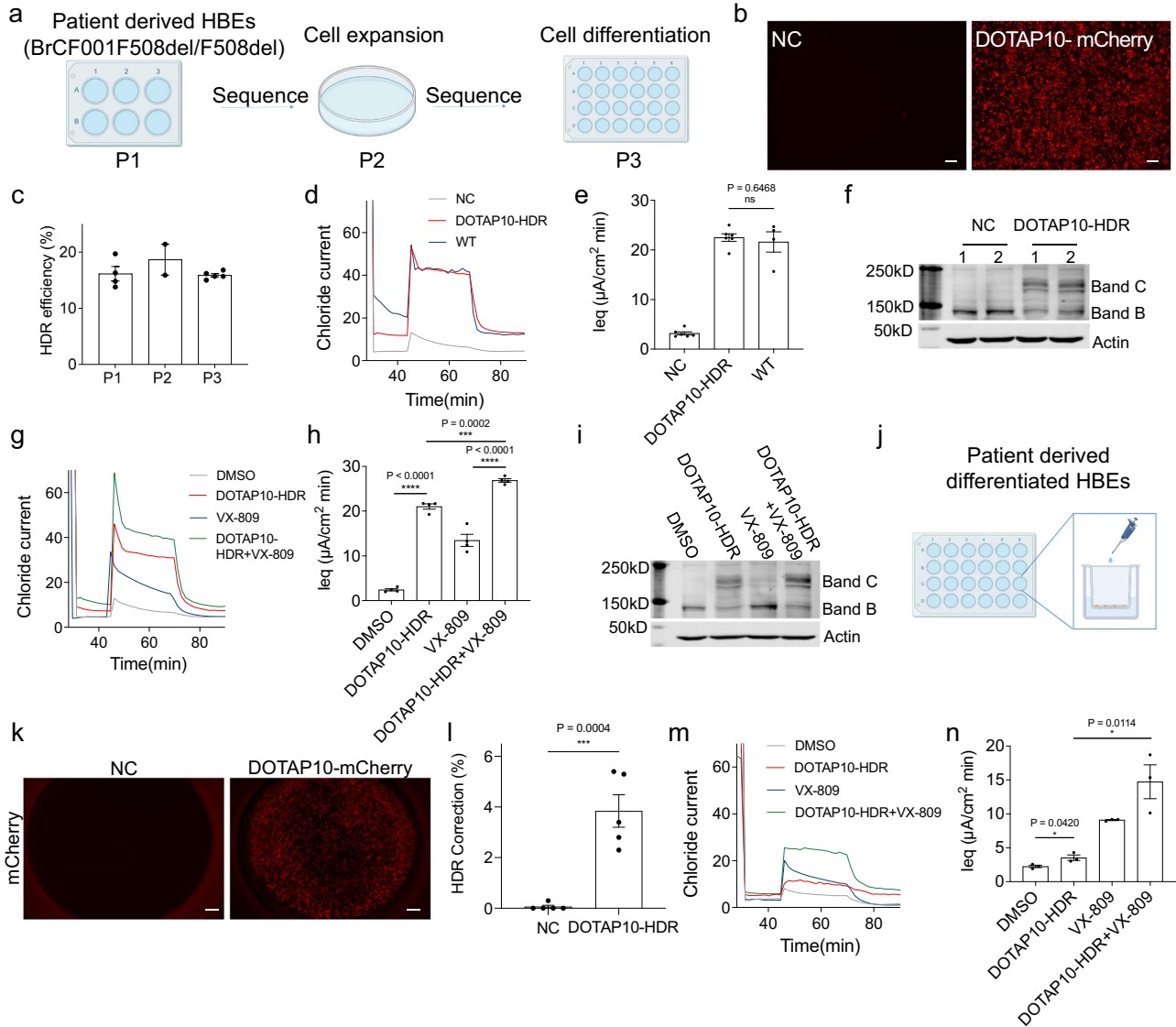

**Fig. 5 | DOTAP LNPs successfully corrected *CFTR* mutation and restored CFTR function in patient-derived human bronchial epithelial (HBE) cells with F508del mutation. a** Undifferentiated HBE cells with F508del mutation (passage 1, P1) were treated with DOTAP10-HDR for 4 days before cell expansion (passage 2, P2). One week after cell expansion, cells were established on transwell inserts for cell differentiation (passage 3, P3). The untreated group was used as negative control (NC) and HBE cells from a healthy donor expressing wild-type (WT) CFTR was used as positive control. **b** DOTAP10 LNPs encapsulating mCherry mRNA (DOTAP10-mCherry) were efficiently internalized into P1 HBE cells, showing bright mCherry fluorescence. Scale bar: 500 µm. **c**, The HDR correction efficiency maintained at as high as 16% after a single DOTAP10-HDR treatment, as evaluated in P1, P2, and P3 HBE cells. DOTAP10-HDR treatment successfully restored CFTR activity, compared to WT CFTR activity (**d**), as confirmed by area under the curve (AUC) analysis (**e**). Data are shown as mean ± s.e.m. (*n* = 4–6 biologically independent samples). **f** Expression of CFTR protein was detected after treatment of DOTAP10-HDR (1, 2 means repeats). Co-treatment of DOTAP10-HDR and VX-809 enhanced Cl⁻

transport and improved CFTR function (**g**), with corresponding AUC values (**h**). There was a significant difference between VX-809 and DOTAP10-HDR + VX-809 groups with a *P* value < 0.0001. Data are shown as mean ± s.e.m. (*n* = 4 biologically independent samples). **i** Increased expression of CFTR protein was detected after co-treatment of DOTAP10-HDR + VX-809. j, DOTAP10-HDR were directly added onto the surface of differentiated HBE cells. **k** DOTAP10-mCherry was efficiently internalized into fully differentiated HBE cells, resulting in bright mCherry fluorescence. Scale bar: 500 µm. **l** DOTAP10-HDR successfully corrected F508del mutation in differentiated HBE cells. Data are shown as mean ± s.e.m. (*n* = 5 biologically independent samples). **m** Co-treatment of DOTAP10-HDR and VX-809 significantly enhanced Cl⁻ transport and improved CFTR function in differentiated HBE cells, as confirmed with corresponding AUC values (*P* = 0.0114) (**n**). Data are shown as mean ± s.e.m. (*n* = 3 biologically independent samples). Two-tailed unpaired *t*-tests were used to determine the significance (*\**P*< 0.05; \*\**P*< 0.01; \*\*\**P*< 0.001; \*\*\*\**P*< 0.0001). Data of (**b**, **k**) were repeated three times independently with similar results.

(DOTAP10-HDR) for 4 days before cell expansion (passage 2, P2). One week after cell expansion, these cells were transferred to transwell inserts for cell differentiation (passage 3, P3) (Fig. 5a). During cell differentiation processing, the HDR correction efficiency was maintained at as high as 16% with DOTAP10-HDR treatment (Fig. 5c). At the end of the study, we evaluated Cl⁻ channel activity mediated by DOTAP10-HDR in differentiated HBE cells (P3). We found that DOTAP10-HDR treatment completely restored CFTR mediated Cl⁻ transport (AUC

22.5 µA/cm2 * min) to the same level as compared to WT HBEs presumed to have 100% CFTR activity (AUC 20.6 µA/cm² * min) (Figs. 5d, e). The western blot result further confirmed the expression of CFTR protein after gene correction by DOTAP10-HDR, while no CFTR protein (at band C) was detected with NC group (Fig. 5f). Meanwhile, we also compared HDR efficacy of DOTAP10-HDR, VX-809 (a CFTR modulator) and their co-treatment on restoration of CFTR function. We found that DOTAP10-HDR showed better effect than VX-809 (AUC 10.9 µA/

cm$^2$*min) (Fig. 5g). Interestingly, co-treatment of DOTAP10-HDR and VX-809 further enhanced Cl$^-$ transport activity (AUC 25.9 μA/cm$^2$*min), suggesting a possible advantage of combinational therapy in future CF studies (Fig. 5h). These results were further confirmed by increased expression of CFTR protein after single treatment of DOTAP10-HDR and co-treatment of DOTAP10-HDR/VX-809 (Fig. 5i).

Compared to undifferentiated HBEs, fully differentiated HBE cells with F508del mutations secrete thicker mucus on the air interface, which is a more challenging barrier for LNP transport. Additionally, these cells have reached a steady state with slow cell division that likely hinders CRISPR/Cas HDR correction that requires cell division. Nevertheless, this model can be a useful addition to broader consideration of in vitro, ex vivo, and in vivo genome correction efforts (Fig. 5j). To pursue this goal, we first examined DOTAP10 LNP delivery of mCherry mRNA to fully differentiated F508del HBE cells. Bright red fluorescence was observed after treatment (Fig. 5k), indicating that DOTAP10 LNPs can transfect fully differentiated HBE cells. Next, DOTAP10-HDR LNPs delivering CRISPR/Cas components were directly added onto the apical surface of differentiated HBE cells with F508del mutation in transwell inserts. Around 4% of HDR correction was detected after three days based on DNA sequencing and TIDER analysis (Fig. 5l) and no toxicity was observed based on transepithelial resistance measurement (Supplementary Fig. 14). Although lower than the correction rate observed in undifferentiated cells, we hypothesize that this is due to cell cycle states of slowly dividing cells that restrict HDR. Similar to what we have observed in undifferentiated HBE cells, co-treatment of DOTAP10-HDR and VX-809 significantly enhanced Cl$^-$ transport activity (AUC 14.7 μA/cm$^2$*min, 71% CFTR functional restoration) and improved CFTR function in differentiated F508del HBE cells (Figs. 5m, n). These results demonstrated that DOTAP LNPs were able to effectively transport through mucus, internalize into HBE cells, and release encapsulated cargoes, providing potential for future efforts in genome editing of CF mutations.

In addition, we evaluated the transfection efficiency of DOTAP10 LNPs delivering tdTomato mRNA (DOTAP10-tdTOM) in differentiated HBE cells via apical and basolateral administration methods (Supplementary Fig. 15). Flow cytometry results showed that DOTAP10-tdTOM transfected >20% of basal cells from the basolateral side, a higher fraction than that from the apical side. Basolateral delivery to ALI cultures may recapitulate better in vivo systemic delivery to the lungs since LNPs reach the differentiated epithelium through systemic IV administration (blood side). Therefore, we hypothesize that gene editors delivered by Lung SORT LNP following systemic administration might achieve higher editing in lung basal cells.

## Discussion

Development of Lung SORT LNPs capable of correcting mutated *CFTR* genes in mouse models of CF and human CF HBEs represent key advances on the path to genome correction therapies. In this paper, we optimized second-generation Lung SORT LNPs (DOTAP40 LNPs), which significantly enhanced lung-targeting efficacy and tolerability for repeat dosing through screening a series of permanently cationic lipids and adjusting their internal ratios among lipid components. Compared to other nanoparticle systems for lung delivery, Lung SORT LNPs access the lungs systemically and do not rely on local intra-tracheal injection[39] or inhalation[40] administration methods. This is beneficial for many lung diseases such as CF since lung diseases are commonly associated with the accumulation of thick and sticky mucus, which adds an additional barrier to local lung delivery.

In this proof-of-concept study, we found that DOTAP40 LNPs could deliver mRNA to lung basal cells with high potency. Apart from a recent study showing highly promising results of peptide nucleic acid (PNA) and donor ssDNA delivery to lung basal cells using PLGA-based nanoparticles for CFTR correction[41], few studies to date have been reported using synthetic carriers to deliver CRISPR/Cas system to achieve effective gene correction in CF disease models. It is important to note certain limitations of the current study. The G542X CF mouse model used in our study does contain the G542X mutations within the genome; however, it does not exhibit pronounced pathological features of CF in the lungs. Also, successful delivery to basal cells in one model does not necessarily translate to another gene target and animal model. Although correction of *CFTR* in various airway cell types is expected to transiently restore CFTR activity, it is now understood that correction of airway stem cells (progenitor cells) would likely be required to enable long-lasting expression of CFTR in epithelial cells derived from progenitor cells. Direct, in vivo editing of these stem cell populations represents the best opportunity for durable "cures" of CF. It remains unknown how long genome edits would persist in the lungs. While the successful functional delivery of mRNA and CRISPR/Cas to mouse lung cells is promising, pivotal work will have to be conducted in the future to understand the feasibility of long-term editing and explore the performance of other editor strategies[42,43] in CF animal models.

The approval of Trikafta[44] has been bittersweet – although the majority of CF patients now have a medicine to take, those with non-sense mutations do not. Since G542X is the most common nonsense mutation of *CFTR* gene, we thus here selected this mutation for in vivo gene correction. We demonstrated that SORT LNPs enabled successful gene correction in mouse lungs of CF model harboring the G542X mutation. Although G542X mouse model does not present CF-related pathological features in mouse lungs, we were able to validate the efficacy of the CRISPR/Cas9 HDR editing system and intercellular delivery and CFTR function restoration through DOTAP LNP mediated gene correction using an ex vivo G542X mouse-derived intestinal organoid model. In addition to lung cells, DOTAP-LNPs can also be used for multi-cargo delivery to multiple cell types and may possess utility in diverse cellular contexts. In addition to G542X mutation, SORT LNPs can also be extended to other mutations, both rare and common. Achieving organ-specific editing will open up many new opportunities for therapeutic intervention.

Moreover, we demonstrated that SORT LNPs successfully corrected *CFTR* and restored CFTR function in patient-derived F508del HBE cells. These trans-well function assays have proven to be highly predictive of clinical benefit and have supported the approval of CFTR modulators that are now used to treat CF patients that harbor specific *CFTR* mutations such as F508del or G551D. The success of our LNP system in this cell model indicates its high potential for future clinical translation.

In summary, we optimized and improved Lung SORT LNP delivery to lung basal cells, and achieved gene correction of G542X and F508del mutations in relevant CF models on the preclinical development path using the CRISPR/Cas9 technology. Furthermore, our research contributes to the diversification of strategies for intracellular gene editor delivery across various cellular contexts. In addition to the recently reported methods of PLGA-based PNA delivery[41], peptide-mediated ribonucleoproteins delivery[45], and endogenous encapsidation for cellular delivery[46], our work expands the existing toolbox in this domain. We acknowledge that additional editing techniques, such as base editing[43], prime editing[47], peptide nucleic acid triplex-forming oligonucleotides, and gene insertion technologies[48] also hold promise for the correction of *CFTR*. In ongoing work, we are utilizing alternative genome editors that do not require double-strand breaks for correction, which we anticipate will increase correction rates in vitro and in vivo. Further clinical development of effective SORT LNPs to deliver CRISPR/Cas to CF-relevant organs in vivo is required and could become a viable path to durable CF therapy.

## Methods
### Ethical statement
Animal experiments were performed according to a protocol approved by the Institution Animal Care and Use Committee of The

University of Texas Southwestern Medical Center and The Institutional Animal Care and Use Committee of Case Western Reserve University (#2014-0064; February 2023). All animals were maintained on a 12-h light, 12-h dark schedule at a mean ambient temperature of 22 °C and humidity of 35–60%. Mice and housed in standard polysulfone microisolator cages in ventilated units with corncob bedding.

## Animals

C57BL/6 mice were obtained from the UTSW Mouse Breeding Core Facility. B6.Cg-Gt(ROSA)26Sortm9(CAG-tdTomato)Hze/J mice (also known as Ai9 mice, JAX:007909) and B6.Cg-Gt(ROSA)26Sortm14(CAG-tdTomato)Hze/J mice (also known as Ai14 mice, JAX:007914) were purchased from The Jackson Laboratory and bred to maintain homozygous expression of the Cre reporter allele that has a loxP-flanked STOP cassette preventing transcription of a CAG promoter-driven red fluorescent tdTomato protein. Following Cre-mediated recombination or Cas9/sgTOM1-mediated deletion, tdTOM reporter mice will express tdTomato fluorescence. Both male and female mice were used for in vivo experiments.

The creation of the G542X mouse model was previously described[28]. Mice homozygous for these mutations were created by breeding heterozygous males and females. Genotyping was completed by PCR analysis using DNA extracts from ear biopsies. To detect the G542X allele (319 bp) primers P1 (5′- ACAAGACAACACAGTTCTCT −3′) and P2 (5′ TCCATGCACCATAACAACAAGT −3′) were used. To detect the wildtype (WT) allele (319 bp) P2 and P3 (5′- ACAAGACAACACAGTTCTTG −3′) were used in a separate reaction. PCR reactions were completed for 40 cycles of 95 °C for 30 s, 58 °C for 30 s and 72 °C for 30 s, and products were run out on 2% agarose gels. All mice were allowed unrestricted access to water and solid chow (Harlan Teklad 7960; Harlan Teklad Global Diets).

## Materials

5A2-SC8 were synthesized and purified by following published protocols[18,20]. 1,2-dioleoyl-sn-glycero-3-phosphoethanolamine (DOPE), 1,2-dioleoyl-3-trimethylammonium-propane (DOTAP), dimethyldioctadecylammonium (DDAB), 1,2-dimyristoyl-sn-glycero-3-ethylphosphocholine (EPC), 1,2-di-O-octadecenyl-3 trimethylammonium propane (chloride salt) (DOTMA), and N1-[2-((1 S)-1-[(3-aminopropyl) amino]-4-[di(3-amino-propyl)amino]butylcarboxamido)ethyl]-3,4-di[oleyloxy]-benzamide (MVL5) were purchased from Avanti Polar Lipids. Cholesterol was purchased from Sigma-Aldrich. 1,2-Dimyristoyl-sn-glycerol-methoxy(poly((ethylene glycol) MW 2000)) (DMG-PEG2000) was purchased from NOF America Corporation. Pur-A-Lyzer Midi Dialysis Kits (WMCO, 3.5 kDa) were purchased from Sigma-Aldrich. Lab-Tek chambered cover glass units were purchased from Thermo Fisher Scientific. Luc mRNA and mCherry mRNA were purchased from TriLink BioTechnologies. Cre mRNA and Cas9 mRNA were produced using in vitro transcription (IVT). tdTomato mRNA was provided by ReCode Therapeutics. D-Luciferin (sodium salt) was purchased from Gold Biotechnology. Anti-p75 NGF Receptor antibody (ab8875) and Donkey Anti-Rabbit IgG H&L (Alexa Fluor® 647) (ab150075) were purchased from Abcam. Anti-CFTR antibody (UNC 596) was purchased from the University of North Carolina and anti-Actin antibody (MAB1501) was purchased from Millipore. Anti-CD45 (Alexa fluor 647, 103124), Anti-EpCam (Alexa fluor 647, 324212), and Anti-CD31 (Pacific Blue, 102422) were purchased from BioLegend. PE-Cy7 anti-human CD271/NGFR (562122) was purchased from BD Biosciences. VX-809 (S1565) was purchased from Selleckchem. End-modified sgRNAs (Supplementary Table 2) were purchased from Synthego. All end-modified HDR templates (Supplementary Table 3) and all primers (Supplementary Table 4–6) were synthesized by Integrated DNA Technologies (IDT).

## LNP formulations

5A2-SC8, DOPE, cholesterol, DMG-PEG, and permanently cationic lipids (DOTAP, DDAB, DOTMA, EPC, or MVL5) were dissolved in ethanol at given molar ratios. mRNA or total nucleic acids (NA, including Cas9 mRNA, sgRNA, and HDR template) were dissolved in Citrate buffer (pH 4). Then the acidic buffer was pipette mixed rapidly into the lipid solution in ethanol at a volume ratio of 3:1, whereas the weight ratio between total lipids and total NA was at 40:1 or 20:1. Afterward, the solution was incubated for 15 min a room temperature to finish assembly. The fresh formulations were directly characterized and used for in vitro assays. For animal experiments, the formulations were dialyzed (Pur-A-Lyzer Midi Dialysis Kits, WMCO 3.5 kDa) against 1× PBS for 3 h to remove acidic buffer and ethanol before in vivo injections. The particle size, polydispersity index, and zeta potential of LNPs were measured using dynamic light scattering (DLS) (Zetasizer Nano ZS machine), and the RNA encapsulation efficiency was measured using Quant-iT RiboGreen RNA assay (Invitrogen, R11491) following published protocols[49].

## In vitro delivery of Luc mRNA and cell viability study

Hela cells were originally obtained from ATCC. Hela cells were seeded into white 96-well plates at a density of $1 \times 10^4$ cells per well the day before transfection. The culture was treated with 5-30% of DOTAP incorporated LNP formulations containing Luc mRNA (20 ng of Luc mRNA per well). ONE-Glo + Tox kits purchased from Promega were used to quantify luciferase expression and evaluate cytotoxicity 24 h after the treatment by following Promega's standard protocol.

## In vitro gene correction in HEK293 cells with Y66H GFP mutation

BFP/GFP HEK293 cells were obtained from the laboratory of Professor Jacob Corn (ETH Zurich). HEK293 cells with Y66H GFP mutation were cultured in DMEM containing 10% FBS and 1% penicillin/streptomycin at 37 °C / 5% CO2. Briefly, HEK293 cells were seeded into 12-well plates at a cell density of $1.5 \times 10^5$ cells per well and incubated overnight. Then, the medium was replaced with 0.5 mL of fresh completed DMEM, and 100 μL of DOTAP10 LNPs with different weight ratios of Cas9 mRNA/sgRNA/HDR template were added (total nucleic acid at 0.8 ng μL$^{-1}$). Two days later, 1 mL of fresh medium was added into each well, to maintain enough nutrition. After three days of treatment, cells of each well were collected, washed, and lysed using 50 μL of 1×passive lysis buffer (Promega) containing 2 μL of proteinase K (Thermofisher), through a PCR program (65 °C for 15 min, 95 °C for 10 min). Afterward, the gene correction sequence was amplified using the following PCR amplification program (95 °C for 5 min; (95 °C for 30 s; 64 °C for 30 s; 72 °C for 30 s) for 35 cycles; 72 °C for 7 min and then keep at 4 °C). Cell lysates were used as DNA templates. PCR amplicons were then purified using PCR purification kits/gel purification kit. Purified PCR amplicons were then sequenced by The McDermott Center Sequencing core facility in UT Southwestern Medical Center. The sequencing data was analyzed using the TIDER web tool (http://shinyapps.datacurators.nl/tider/), to calculate HDR correction efficiency.

## Confocal Imaging of HEK293 cells with Y66H GFP mutation

$6 \times 10^4$ HEK293 cells with Y66H GFP mutation were seeded in 8-well confocal dish overnight. Afterward, DOTAP10-HDR LNPs encapsulating Cas9 mRNA, sgRNA, and HDR template (0.5:1:6, weight ratio) were added into cells and incubated for two days. NC group (PBS treatment) and DOTAP10-NHEJ LNPs encapsulating Cas9 mRNA and sgRNA group were used as controls. The dose of total nucleic acid per well was 0.4 ng μL$^{-1}$. After two days, each well was stained with a cell mask deep red stain solution (1:1000 dilution) at 37 °C for 10 min and washed carefully 3 times with PBS buffer. Then cells were observed using confocal microscopy (Zeiss LSM 700).

## In vivo Luc mRNA delivery

C57BL/6 mice with weight at around 20 g were i.v. injected with various Luc mRNA formulations (0.1 mg kg⁻¹ of Luc mRNA, $n=3$ per group). After 6 h, these mice were injected with D-Luciferin (150 mg kg⁻¹, intraperitoneal) and imaged immediately using an IVIS Lumina system (Perkin Elmer). Afterward, these mice were killed, and tissues were excised for ex vivo imaging using IVIS Lumina system.

## In vivo toxicity evaluation

C57BL/6 mice with weights at 18–20 g, were randomly divided into four groups: PBS group, DDAB30 LNPs group, DOTAP40 LNPs group, and Lipopolysaccharide (LPS) group ($n=5$ per group). Briefly, DDAB30 LNPs and DOTAP40 LNPs were injected into a mouse tail veil, with mCherry mRNA dose at 1.5 mg kg⁻¹ (total lipid/total RNA at 40:1). PBS treated group was used as negative control. Intraperitoneal injection (I.P.) of LPS (5 mg kg⁻¹) was used as a positive control. After treatment of 24 h, the whole blood was collected, and the serum was separated to measure the liver function (AST and ALT) and kidney function (BUN and CREA) using the UT Southwestern Metabolic Phenotyping Core. Mice in LPS group were sacrificed after treated for two days and other groups were sacrificed at day 5. During the period, body weight changes of these mice were monitored. On day 5, mice were sacrificed and the tissue weights (liver, spleen, and lung) of each mouse were recorded. All tissue (heart, liver, spleen, lung, and kidney) sections with H&E staining were then prepared and analyzed at the UT Southwestern Tissue Management shared resource. Slides were scanned using NANOZOOMER (NDP.scan software version 3.1.9, Hamamatsu) and analyzed using NDP.view 2 software (version 2.7.25, Hamamatsu).

## In vivo repeat dosing of DOTAP40-Luc

C57BL/6 mice with weight at around 20 g were i.v. injected with DOTAP-40-Luc (0.1 mg kg⁻¹ Luc mRNA, 20:1) once a week, three weeks in total. Mice were injected with D-Luciferin (150 mg kg⁻¹, intraperitoneal) 6 h after each injection and imaged immediately using an AMI-HTX (Spectral Instruments Imaging) system. Following the in vivo imaging after the third injection, these mice were sacrificed and dissected to collect tissues for ex vivo imaging using an AMI-HTX (Spectral Instruments Imaging) system. The images were processed with the Aura software v.4.0.7 (Spectral Instruments Imaging).

## Off-organ editing evaluation using Ai14 tdTOM reporter mice

To evaluate in vivo off-organ editing, Ai14 tdTOM reporter mice of comparable weight were utilized. The formulations were prepared as previously described, with a total lipid to total nucleic acid weight ratio of 20:1. Ai14 reporter mice were treated with DOTAP40-NHEJ (Cas9 mRNA:sgTOM1 = 2:1, wt/wt) at 2 mg kg⁻¹ total RNA dosage by intravenous administrations with the same doing regime used for in vivo correction experiment with G542X mice (once a week, three injections in total). Three weeks following the third injection, mice ($n=3$ per group) were sacrificed, and different organs (brain, heart, lung, liver, pancreas, spleen, kidney, GI tract, and reproductive organs) were collected for ex vivo imaging to determine in vivo editing-mediated tdTOM fluorescence increase compared to the untreated control mice using an AMI-HTX (Spectral Instruments Imaging) system. The images were processed with the Aura software v.4.0.7 (Spectral Instruments Imaging).

## In vivo gene editing (Cas9 mRNA/sgRNA and Cas9/sgRNA/ssDNA) in the tdTOM mouse lungs

To evaluate in vivo gene editing by using a two-component system (Cas9 mRNA/sgRNA) against a three-component system (Cas9 mRNA/sgRNA/ssDNA), Ai14 tdTOM reporter mice (B6.Cg-Gt(ROSA)26Sortm14(CAG-tdTomato)Hze/J mice, JAX:007914) of comparable weight and same-sex were selected. The formulations were prepared as previously described, with a total lipid to total nucleic acid weight

ratio of 20:1. tdTOM mice were treated once a week with DOTAP40-NHEJ (Cas9 mRNA:sgTOM1 = 2:1, wt/wt) and DOTAP40-HDR (Cas9 mRNA: sgTOM1: ssDNA HDR template for CF G542X = 2:1:3, wt/wt/wt) formulations respectively with a total RNA dosage of 1 mg kg⁻¹ by intravenous administrations. One week following the third injection, mice ($n=3$ per group) were sacrificed, and the lung cells were isolated and stained for cytometry analysis to determine the proportion of tdTOM+ cells in each different cell type. Briefly, mouse lungs were resected, and the tissues were placed into ice-cold PBS. The lung tissue was then cut into small pieces and transferred into a 50 mL tube containing 10 mL of 1× lung digestion media [RPMI dissociation medium (1:1 vol/vol) RPMI supplemented with 2% wt/vol BSA, 300 U/mL collagenase, 100 U/mL hyaluronidase]. The 50 mL tube was then incubated at 37 °C for 1 hr while shaking at 180 rpm. After incubation, the homogenized lung cell solution was pipetted up and down several times to remove cell clumps and finally filtered through a 70-micron cell strainer into a new 50 mL falcon tube. The filter was washed with 10 mL wash buffer consisting of cold PBS and 2% fetal bovine serum (FBS). The sample was then centrifuged at 900×g for 5 min. The supernatant was removed, and the cell pellet was resuspended in 10 ml of cold wash buffer. Next, the red blood cells were lysed by resuspending the cell pellet in 5 mL of 1X RBC lysis buffer (BioLegend) at room temperature for 5 min. After 5 min, 10 mL of wash media was added to the sample. The sample was centrifuged at 900×g for 5 min. Finally, the RBC free cell pellet was resuspended in 5 mL of cell staining buffer (BioLegend) and proceeded with antibody staining for flow cytometry.

Single-cell suspensions obtained from the mouse lungs were pre-blocked with a mouse Fc-receptor blocker (BioLegend) for 15 min. Subsequently, cells were labeled with an Alexa fluor 488-conjugated anti-mouse CD31, Pacific, blue-conjugated anti-mouse CD45 and Alexa fluor 647 -conjugated anti-mouse EpCAM antibodies (all from BioLegend) by incubating 100 μL of cell suspension with antibodies for 15 min on ice. Ghost dye red (BioLegend) was used to identify the dead cells. Next, the cell pellet was washed 3-times with cell staining buffer to remove excess antibodies. Finally, the cell pellet was resuspended in 500 μL of cold cell staining buffer and kept on ice until analysis by flow cytometer. Cells were then analyzed by Becton Dickenson (BD) LSR Fortessa flow cytometer. Data obtained from the flow cytometer were finally analyzed by flowjo software (BD).

## Basal stem cells editing in tdTOM mice with DOTAP40-Cre LNP treatment

tdTOM mice were treated with DOTAP40 LNPs encapsulating Cre recombinase mRNA twice by I.V. injection (2 mg kg⁻¹ Cre mRNA, total lipid/total NA at 20:1, $n=4$). Two days after the second injection, mouse lungs were collected, and lung cells were isolated and stained. Briefly, lung tissue was minced using A blade and transferred to an EP tube containing 250 μL of 2×digestion medium (90 units μL⁻¹ collagenase I, 50 units μL⁻¹ DNase I, and 60 units μL⁻¹ hyaluronidase, 1% FBS) and homogenized with tissue grinder. Afterwards, the tissue solution was transferred to a 15 mL centrifuge tube containing 10 mL of 2×digestion medium and incubated at 37 °C for 1 h with shaking. Next, the lung solution was filtered using a 70-μm filter and washed once with 1× PBS. A cell pellet was obtained by centrifuging for 10 min at a speed of 500×g at 4 °C. The supernatant was removed, and the cell pellet was resuspended in 2 mL of 1×red blood cell lysis buffer (BioLegend, 420301) and incubated on ice for 5 min. After incubation, 4 mL of cell staining buffer (BioLegend) was added to stop red blood cell lysis. The solution was then centrifuged again at 500×g for 10 min to obtain a cell pellet. The single cells were resuspended in cell staining buffer to make a cell solution at a density of $1–5 × 10^6$ cells mL⁻¹. 100 uL of cell solution was transferred to a new EP tube to incubate with Anti-p75 NGF receptor antibody (1:100 dilution) for 30 mins on ice, to stain lung basal cells. Afterwards, the cells were washed with 1× PBS buffer twice

and incubated with Donkey Anti-Rabbit IgG H&L (Alexa Fluor® 647) (1:2000) for 30 mins on ice, protecting from light (total volume 100 μL). The stained cells were washed twice with 1 mL of 1× PBS, then resuspended in 500 μL 1× PBS for flow cytometry analysis using a LSRForessa SORP (version 8.0.1, BD Biosciences) in the Moody Foundation Flow Cytometry Facility. Ghost Dye Red 780 (Tonbo Biosciences, 13-0865-T500) was used to discriminate live cells. Nanoparticle only treatment group (NC) was used as negative control. The data of flow cytometry were analyzed using FLOWJO software version 7.6 (FLOWJO).

### Gene correction in CF mouse model with G542X mutation

DOTAP40-HDR LNPs delivering Cas9 mRNA, sgRNA, and HDR template were I.V. injected into CF mouse model containing G542X mutation every week for a total of three times (total NA at 2 mg kg$^{-1}$, total lipid/total NA at 20:1, $n = 6$). One week after the last injection, mouse whole lungs were collected and homogenized for genome DNA extraction. Using genome DNA as DNA template, the gene correction sequence was amplified using a PCR amplification program (98 °C for 2 min; (98 °C for 10 s; 53 °C for 30 s; 72 °C for 30 s) for 25 cycles; 72 °C for 2 min and then keep at 4 °C). A second round of PCR was used to add Illumina flow cell binding sequences and barcodes. PCR amplicons were then purified using PCR purification kits (Qiagen) and quantified by Qubit dsDNA high-sensitivity assay (Invitrogen) before pooling and loading on an Illumina MiSeq. Deep sequencing reads were demultiplexed and analyzed with CRISPResso2 (https://crispresso.pinellolab.partners.org/) for HDR correction frequency. NP only treatment group was used as negative control (NC).

### TA Cloning to validate gene correction event in DOTAP40-HDR treated mice lung

Taq polymerase-amplified PCR products of gDNA extracted from DOTAP40-HDR treated G542X mice lungs were inserted into a plasmid by using the TOPO® TA Cloning® Kits following the standard protocol. 50 single colonies were picked and cultured overnight in LB medium containing 50 ug/mL ampicillin. Plasmid DNA from each single colony was extracted using QIAprep Spin Miniprep Kit and later sequenced by The McDermott Center Sequencing core facility in UT Southwestern Medical Center to detect DOTAP40-HDR mediated gene correction events.

### Off-target analyses

Candidate off-target sites were predicted with CRISPOR and the top six sites by the MIT Specificity score[1,2]. Genomic DNA extracted from DOTAP40-HDR treated G542X mouse lungs were used as DNA template for PCR amplification at those predicted off-target sites using primers listed in the supplementary Table S5 with additional 8 bp barcodes on both ends. 40 ng of gDNA was used for the PCR reaction using a Phusion Plus Green PCR Master Mix (Thermo Fisher) with an amplification program (98 °C for 3 min; (98 °C for 10 s; 60 °C for 30 s; 72 °C for 30 s) for 30 cycles; 72 °C for 2 min). PCR amplicons were then purified using PCR purification kits (Qiagen) and quantified by Qubit dsDNA high-sensitivity assay (Invitrogen) amplicon deep-sequencing library was then prepared and later sequenced by Novogene using Illumina NovaSeq 6000. After demultiplexing, amplicon sequencing data were analyzed with CRISPResso2[3] (https://crispresso.pinellolab.partners.org/) to determine the editing efficiency.

### Crypt harvest and intestinal organoid culture

Intestinal organoids were cultured similarly to previously described methods[28]. Mice were sacrificed by $CO_2$ asphyxiation, and 20 cm of intestine measured from the stomach were removed. Fecal matter was flushed from the intestine with $Ca^{2+}$ and $Mg^{2+}$-free PBS, and the intestine was flayed using dissecting scissors. The villi were scraped from the small intestine using a microscope slide, and the intestine was cut into ~1 cm segments, which were suspended in 2 mM EDTA in $Ca^{2+}$ and $Mg^{2+}$-free PBS. The intestinal segments were incubated on a shaker for 30 min at room temperature. The segments were then vortexed at for 10 s, allowed to settle, and then the supernatant was removed and stored in a 10 cm dish. This process was repeated until four supernatant fractions were produced. The fractions were inspected under a microscope, and the fraction that was most enriched for crypts was passed through a 70 μm cell strainer. The crypts were pelleted at 1,000xG for 10 min, then resuspended in 1:1 mixture of Intesticult Organoid Growth Media (OGM; STEMCELL Technologies) and MatriGel (Corning) at a concentration of 10 crypts/μl. The organoids were seeded to 12-well plates, with 70 μl Matrigel:OGM added to each well in 4-5 droplets. The plate was placed in a 37 °C/5% $CO_2$ incubator for 15 min to allow the MatriGel to harden. The MatriGel domes were then immersed in 1 mL OGM and returned to the 37 °C/5% $CO_2$ incubator. OGM was changed every 3–4 days, and the organoids were passaged once every 5–7 days.

### Ex vivo G542X correction in intestinal organoids using DOTAP LNPs

Organoids were grown in Matrigel droplets in a 12 well plate to approximately 75% confluency with Mouse Intesticult OGM containing 10 uM Y-27632 (STEMCELL Technologies) and 5uM CHIR 99021 (Sigma-Aldrich). Organoids were released from Matrigel using PBS and centrifugation. Pelleted organoids are resuspended in 1 ml of Acutase (Life Technologies) and incubated at 37 °C for 5 min to digest organoids into single cells. Digestion was halted by quenching the Acutase with 2 ml of DMEM media containing 10% FBS. Cells were then placed in an eppedorf tube with 200 μL of OGM containing (2.5 ng/μL of total NA) and incubated at 37 °C/5% $CO_2$ for 4 h. Cells and media were then placed in one well of a Matrigel-coated 96-well plate. Cells were grown at 37 °C/5% $CO_2$ for approximately 4–5 days until full organoids developed from the surviving single stem cells. Forskolin-induced swelling of the resulting intestinal organoids were carried out as previously described[28,34] with small modifications. Forskolin was added to each will with a final concentration of 10 μM. Kinetic brightfield images of FIS were acquired under live cell conditions with a Lionheart FX Automated Microscope (Agilent). After one hour of FIS, each organoid was scored as either corrected for CFTR activity, if the organoid swelled, or not corrected for CFTR activity, if the organoid did not swell. 8 96-wells per treatment group were used for each experiment. DNA was isolated from each well for sequencing of the G542X locus.

### Uptake of DOTAP10-mCherry in undifferentiated HBE cells

Cystic fibrosis patient-derived undifferentiated HBE cells with homozygous F508del mutation were provided by Dr. Hillary Valley and the Cystic Fibrosis Foundation Therapeutic Lab. The undifferentiated HBE cells with F508del mutation were thawed and seeded at 150 K/well on 6-well plates. Cultures were maintained every day in Lonza BEGM for 4 days prior to treatment with DOTAP10 LNPs encapsulating mCherry mRNA (DOTAP10-mCherry) for 24 h (mCherry mRNA dose at 1 μg well$^{-1}$). Afterwards, the cellular uptake of DOTAP10-mCherry was imaged using a Keyence microscope (BZ-X Analyzer software version 1.0.0, Keyence Corporation).

### LNPs treatment in undifferentiated HBE cells with F508del mutation

Cystic fibrosis patient-derived HBE cells with homozygous F508del mutation (passage 1, P1) were thawed and seeded at 150 K/well on 6-well plates on day 0. Cultures were maintained every day in Lonza BEGM for 4 days prior to treatment. At day 4, P1 cells were treated with DOTAP10-HDR LNPs (at a total NA concentration of 800 ng/well) for 4 days before being transferred into a 10 cm2 dish and T-75 flask for cell expansion (passage 2, P2). Untreated group was used as negative control (NC). One week later, these cells were established on Transwell

Inserts for cell differentiation (passage 3, P3). Cultures were maintained on a 3 days/week feeding routine with Vertex ALI media containing Ultroser G serum for 4 weeks before following function studies. Several wells from P1, P2, and P3 were collected and lysed using 1×passive lysis buffer (Promega) containing proteinase K (Thermofisher), through a PCR program (65 °C for 15 min, 95 °C for 10 min). Afterward, the gene correction sequence was amplified using the following PCR amplification program (95 °C for 5 min; (95 °C for 30 s; 64 °C for 30 s; 72 °C for 40 s) for 35 cycles; 72 °C for 7 min and then keep at 4 °C). Cell lysates were used as DNA templates. PCR amplicons were then purified using a gel purification kit. Purified PCR amplicons were then sequenced by The McDermott Center Sequencing core facility in UT Southwestern Medical Center. The sequencing data was analyzed using the TIDER webtool (http://shinyapps.datacurators.nl/tider/), to calculate HDR correction efficiency.

### LNPs treatment in differentiated HBE cells with F508del mutation

Patient derived HBE cells with homozygous F508del mutation (passage 1, P1) were thawed and seeded at 150 K/well on 6-well plates. Cultures were maintained every day in Lonza BEGM for one week and then transferred to a T-75 flask for cell expansion (passage 2, P2). Cultures were maintained every day in Lonza BEGM for another week. Next, Passage 3 (P3) cultures were established on HTS Transwell Inserts and airlifted; cultures were then maintained on a 3 days/week feeding routine with Vertex ALI media containing Ultroser G serum for cell differentiation. Afterwards, P3 HBEs were washed according to ReCode wash protocol and 12 µg of DOTAP10-HDR formulation was directly administered to apical surface of cultures in liquid bolus and cultured for another four weeks before following function studies. Several wells were collected after treated for 3 days and lysed using 30 µL of 1×passive lysis buffer (Promega) containing 2 µL of proteinase K (Thermofisher), through a PCR program (65 °C for 15 min, 95 °C for 10 min). Afterwards, the gene correction sequence was amplified using the following PCR amplification program (95 °C for 5 min; (95 °C for 30 s; 64 °C for 30 s; 72 °C for 40 s) for 35 cycles; 72 °C for 7 min and then keep at 4 °C). Cell lysates were used as DNA templates. PCR amplicons were then purified using gel purification kit. Purified PCR amplicons were then sequenced by The McDermott Center Sequencing core facility in UT Southwestern Medical Center. The sequencing data was analyzed using the TIDER web tool (http://shinyapps.datacurators.nl/tider/), to calculate HDR correction efficiency.

### Functional studies in differentiated HBE cells

CFTR function was measured as transepithelial chloride secretion across fully differentiated P3 F508del/F508del HBEs using a Transepithelial Current Clamp (TECC) and 24-well electrode manifold (EP Devices). Selected wells were pretreated for 24 h with 3 µM VX-809 or 0.2% DMSO. To prepare the plate for functional analysis, differentiation media was replaced with HEPES buffered F-12 assay medium (pH 7.4) on both the apical and basolateral sides. After a 45 min incubation in a 37 °C incubator without $CO_2$, the 24-well transwell was mounted onto a 37 °C heated platform, and transepithelial resistance (Rt) and voltage (Vt) were continuously measured. Baseline values were measured for ~30 min, then Rt and Vt were measured for ~15 min after apical addition of benzamil (6 µM final concentration), for ~30 min after simultaneous apical/basolateral addition of forskolin (10 uM final concentration)/VX-770 (1 µM final concentration), and for ~15 min after basolateral addition of bumetanide (20 µM final concentration). Benzamil inhibits sodium transport by the epithelial sodium channel, forskolin increases intracellular cAMP which results in the downstream activation of the CFTR channel, VX-770 is a potentiator for the F508del-CFTR channel, and bumetanide inhibits the basolateral membrane NaCl2 cotransporter which ultimately blocks Cl2 transport. CFTR functional results are presented as equivalent chloride current

(Ieq) which was calculated using Ohm's law, $Ieq = Vt/Rt$. Primary HBE cells collected from a healthy donor who has wild-type $CFTR^{wt/wt}$ were used as a positive control.

### Western blot

P3 F508del HBE cells were lysed immediately following TECC-24 functional analysis. After gently washing HBEs with PBS, 35 µl of RIPA buffer (150 mM NaCl, 50 mM Tris·HCl (pH 8), 1% IGEPAL, 0.5% sodium deoxycholate, 0.1% SDS, 1% protease inhibitor) was added directly to transwell insert, rocked for 1 h at 4 °C, then collected in 1.5 ml tubes on ice. Insoluble fractions were separated by centrifugation at 1600 x g for 5 min at 4 °C. The supernatant was then collected in a separate 1.5 ml tube and Laemmli sample buffer was added at 1:6 dilution. Entire lysate was loaded, and protein was separated on a 7-10% hand-cast SDS-PAGE gel at 100 V for 1.5 – 2 h. Next, protein was transferred to an Immobilon-FL PVDF (Millipore, #IPFL00010) membrane for 1 h at 100 V at 4 °C using the Mini Trans-Blot Electrophoretic Transfer Cell system. The membrane was then blocked with 5% milk in TBS for 1 h at room temperature. Anti-CFTR primary antibody was incubated overnight at 1:15,000 dilution in 1% Tween 20 in TBS (TBST) at 4 °C. The following day, anti-actin was added at 1:50,000 dilution for 30 min in TBST at room temperature. The membrane was then washed 3 times with TBST and incubated with IRDye 680RD Goat anti-Mouse IgG secondary antibody (LI-COR, 926-68070) at 1:10,000 dilution in TBST for 1 h at room temperature. The membrane was washed again three times with TBST and once with TBS, and then finally imaged using the LI-COR Odyssey CLx imaging system.

### Uptake of DOTAP10-tdTOM in fully differentiated HBE cells

To test the tdTom mRNA uptake in fully differentiated HBE cells by DOTAP LNP-mediated delivery (DOTAP10-tdTOM), human bronchial epithelial (HBE) cells were treated DOTAP10-tdTOM (12 µg total tdTOM mRNA per well) via either apical side or basolateral side of the transwell. 24 h after a single treatment, cells were detached from transwells, followed by staining and flow cytometry analysis to determine the proportion of tdTOM⁺ cells in basal and differentiated cell populations. To detach the HBE from transwell inserts, differentiation culture media was aspirated from both the apical and basolateral sides of the transwell. Each well was washed on the apical side with 100 µL 1X DPBS before adding 500 µL of 4 °C Accumax solution (Sigma, SCR006) to the basolateral side, and 100 µL of 4 °C Accumax solution to the apical side. Plates were incubated at 37 °C for 30 min, then 100 µL of 4 °C Accumax solution was added to the apical side, gently pipetting up and down on the apical side before putting the plates back in the incubator at 37 °C for 20 min. 100 µL of 4 °C Accumax solution was again added to the apical side, gently pipetting up and down on the apical side before putting the plates back in the incubator at 37 °C for 20 min. A cell pellet was obtained by centrifuging for 5 min at a speed of 1000 g at room temperature and removing the supernatant.

For staining of HBE, FACS buffer was prepared at a final concentration of 5 mM EDTA (Invitrogen, 15575-038), 10 µM Y-27632 (STEMCELL, 72302), 10% FBS in 1X DPBS. Resuspended each cell pellet in 100 µL of FACS buffer with PE-Cy7 anti-human CD271/NGFR (BD, 562122) at 1:100 dilution. Cells were incubated for 20 min in the dark at 4 °C. The stained cells were washed twice with 1 mL 1X DPBS, then resuspended in 500 µL FACS buffer with SYTOX Blue (ThermoFisher, S34857) at 1:1000 dilution to discriminate live cells. The HBE cells were then analyzed using ThermoFisher Attune CytPix machine (ThermoFisher) and FlowJo software (BD Life Sciences) to assess the DOTAP10-tdTOM uptake.

### Display items

The images of lipid nanoparticles, mice, syringes, cells, plates, and pipettes (Figs. 1, 3a, f, 4c, f, 5a, j and Supplementary figs. 4a, 10a, b, 13a) were created with BioRender.com.

## Statistical analysis

Statistical analyses were performed using Prism 9.5.1 (GraphPad Software). All data are presented as means ± s.e.m. as indicated. Two-tailed unpaired $t$-test was used to determine the significance of the comparisons of data. $P$ values < 0.05 were considered statistically significant (*$P$ < 0.05; **$P$ < 0.01; ***$P$ < 0.001; ****$P$ < 0.0001).

## Reporting summary

Further information on research design is available in the Nature Portfolio Reporting Summary linked to this article.

## Data availability

All data needed to evaluate the conclusions in the paper are present in the Main Text and Supplementary Material. DNA sequencing files can be accessed at the National Center for Biotechnology Information Sequence Read Archive (NCBI SRA) with accession code PRJNA1012742. Source data are provided in the Source data file. Source data are provided with this paper.

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

## Acknowledgements

We appreciate and thank the individuals who donated their primary bronchial cells; Dr. Hillary Valley and the Cystic Fibrosis Foundation Therapeutic Lab for providing HBE cells and advice; Dr. Domenico Orlando for helping HeLa cell experiments; Dr. Andreas Chai, Dr. Hui Li and Prof. Eric N. Olson for helping with deep sequencing, the UTSW Small Animal Imaging Shared Resource supported in part by National Cancer Institute (P30CA142543); the UTSW McDermott Center Sanger Sequencing Core; the UTSW McDermott Center Next Generation Sequencing Core; the UTSW Metabolic Phenotyping Core; the Moody Foundation Flow Cytometry Facility, and the CF Mouse Model Resource Center at CWRU funded by the CFF (HODGES19R1 to C.A.H.).The research was primarily supported by a grant from the Cystic Fibrosis Foundation (CFF) (SIEGWA18XX0 to D.J.S.). The research was also supported in part by National Institutes of Health (NIH) National Institute of Biomedical Imaging and Bioengineering (NIBIB) (RO1 EBO25192-01A1 to D.J.S.), a Sponsored Research Agreement with ReCode Therapeutics and a CPRIT Training Grant (RP160157 to T.W.).

## Author contributions

T.W., Y.S., Q.C., and D.J.S. designed the research. T.W., Y.S., Q.C., Z.T., C.A.H., L.T.J., S.C., M.L.C., J.W., M.J.T., X.L. X.W., and Y.X., performed the experiments. All the authors were involved in the data analyses. T.W., Y.S., C.A.H., and D.J.S. wrote the manuscript, and all authors discussed and commented on the manuscript.

## Competing interests

A provisional patent application covering compositions, methods, and uses for targeting cystic fibrosis and related disorders has been filed by UT Southwestern naming T.W., Y.S., Q.C. and D.J.S as inventors. D.J.S. discloses financial interests in Signify Bio, ReCode Therapeutics, and Tome Biosciences. D.J.S. is a co-founder and member of the scientific advisory board of ReCode Therapeutics, which has licensed intellectual property from UT Southwestern. The remaining authors declare no competing interests.
