## [Peer Review File · Nature Communications]

Reviewers' Comments:

Reviewer #1:

Remarks to the Author:

Wei et al. Nature Communications Review

This manuscript describes an exciting study of tissue-specific targeting LNPs for gene editing of Cystic Fibrosis (CF). Wei and colleagues apply their previously developed SORT LNP technology to deliver CRISPR/Cas9-based editing agents to correct CF mutations in cell culture and in vivo. While gene editing efficiency in the lungs of G542X mice was somewhat low (~4%) and variable following systemic delivery of lung SORT LNPs, this study serves as a proof of concept for successful intravenous lung delivery of editing agents using LNPs. Additionally, the demonstration of gene editing of CF stop codon mutations is timely, given that patients with these mutations are not eligible for TRIKAFTA and are in need of other treatment options.

Major Criticisms:

- The authors indicate basal stem cells in the lung as the ideal target for editing therapeutics due to the potential for long-lasting expression of CFTR in epithelial cells, and demonstrate that SORT LNPs are able to transfect these stem cells. However, the true test of the above mentioned hypothesis would be an assessment of the longevity of the editing response or CFTR expression in vivo, which is not provided.
- Other recent work, with an alternate nanoparticle delivery system (ref. 40), also provided evidence of delivery to basal cells, which might be a good point of comparison.
- The intestinal organoid studies are interesting and are helpful for testing of the editing agents in tissues from the G542X mouse model but they do not provide evidence of in vivo delivery effectiveness. This seems particularly important in this case, since the in vivo delivered LNPs are designed to target the airways and not the intestines.
- An in vivo assessment of phenotypic correction in the airways would be more compelling to show a correlation with genotypic correction. Others have used nasal potential difference measurements in CF mouse models. An in vivo assay such as this should be included.
- Basolateral as opposed to apical administration of LNPs of ALI cultures would be a better recapitulation of IV delivery.
- There is no mention of off-target effects of Cas9 editing in terms of editing in off-target organs or off-target genomic sites in the lungs. This analysis should be done as a standard assessment of safety and genotoxicity of CRISPR/Cas9-based editing.
- Immunogenicity of LNPs (ex. serum and BALF cytokine levels) should be assessed in addition to the toxicity indicators assessed.
- Some of the claims in the manuscript are vague, and the limitations of the study would be clearer if the authors could be more specific. For example, the claim that basal cells are edited is based on studies with Cre mRNA in a reporter mouse, not CFTR editing in a CF mouse. Successful delivery to basal cells in one model does not necessarily translate to another gene target and animal model.

Minor Criticisms:

- Do lung SORT LNPs reach other airway cells (ex. trachea, nose)? A biodistribution assessment of these tissues would be helpful.
- Why is there a positive HDR editing signal in NC control samples in Figure 4? Shouldn't editing readouts for these controls be closer to 0%?
- Are HeLa cells the best screening cell line to use for in vitro studies if the intended cell type is primary HBEs grown at different culture conditions?
- It would be helpful to note changes, if any, in LNP characteristics (size, surface charge, etc.) and biodistribution for the various formulations incorporating gene editing cargo (mRNA, sgRNA, ssDNA) during optimization of these LNPs.
- Why was a weight ratio used for Cas9 mRNA/sgRNA LNP optimization as opposed to molar ratios? Providing conversions to molar ratios might be helpful for many readers looking at similar approaches. In addition, it would be interesting to know the weight ratio of total nucleic acid:lipid in the final preparations (I believe 40:1 or 20:1 from Fig 4), and estimates of how many copies of each molecule are in each LNP particle.

- How were HDR templates designed/chosen? This is unclear.

Reviewer #2:

Remarks to the Author:

Reviewer #3:

Remarks to the Author:

SIGNIFICANCE:

The development of CFTR modulator compounds has provided a large benefit for roughly 90% of CF patients. However, the remaining 10% do not respond to these modulators and require other therapeutic approaches. While gene correction via CRISPR/Cas would be a significant advance towards such therapies, this approach is currently limited by the lack of efficient genome editor delivery carriers. The current study reports development of improved Lung Selective Organ Targeting (SORT) Lipid Nanoparticles (LNPs) that can efficiently deliver Cas9 mRNA, sgRNA, and donor ssDNA templates for precise homology-directed repair (HDR)-mediated gene correction in ex vivo and in vivo CF models. These new LNPs successfully corrected the G542X CFTR mutation in G542X/G542X mice and in patient-derived F508del/F508del human bronchial epithelial (HBE) cells, leading to restored expression of CFTR protein and chloride transport. This more efficient formulation of LNPs and the demonstration of its utility to promote gene correction is an important development in the CF field.

CRITIQUE:

I think this is a solid manuscript describing the development and utility of a new formulation of LNP for the delivery of gene-editing components. I have the following questions & comments:

- 1) Page 5, top paragraph (and Figure 4c): You chose to do 3 weekly injections of your cocktail and then harvest after one additional week. Did you try any other dosing schedules (shorter or longer)? If not, how do you know this is optimal?
- 2) Page 5, top paragraph (and Figure 4d): You state in the text that you obtained a correction rate for G542X in vivo of 4%. However, figure 4d seems to indicate a rate of 3%. Please address this disparity.
- 3) You suggest that by isolating all lung cells, the rate of editing is likely diluted through isolation of cells assumed to be "irrelevant for CF treatment". If you sorted lung cells using an epithelial cell marker, could you get a better estimate of the efficiency of correction?
- 4) Page 5, line 22: you state that >20% of organoids swelled after treatment, while your HDR correction rate in intestinal organoids was only 2.5% (Figure 4i). Is this because each organoid is made up of a cluster of starting cells? Please explain.
- 5) Figure 5e: What level of correction does this AUC represent? In other words, please indicate the AUC obtained with WT HBE cells for comparison.
- 6) Figure 5h: It is not clear that some of these data are significantly different. In particular, please provide p values comparing DOTAP vs. DOTAP + VX-809. Is there a significant additive effect?
- 7) Figure 5n: It is not clear that DMSO and DOTAP are significantly different, or if VX-809 and DOTAP + VX-809 are significantly different. In addition, if DMSO and DOTAP are the same, then the further increase from VX-809 vs. DOTAP + VX-809 could be an artifact, possibly due to greater cell permeabilization due to the lipids, thus allowing more VX-809 to enter.

Reviewer #4:

Remarks to the Author:

The manuscript by Tuo Wei from Daniel Siegwart group reports on lipid nanoparticles optimized from a former study that are optimized for mRNA delivery of Cas9 together with the sgRNA and a donor ssDNA to perform HDR in lungs. The authors analyze the efficacy of LNPs delivery in vitro

and in mice to test the tissue tropism and editing efficacy in lungs. CFTR functional recovery is tested in vitro using intestinal organoids derived from treated mice and primary human epithelial cells from patients (F508del).

This study stems from a previous work by the same group developing LNP and here they further optimize LNPs for in vivo delivery and HDR editing. The reported method of delivery has obvious implications for the advancement therapeutic strategies for cystic fibrosis.

The work is very well performed and includes convincing in vivo data.

Major comments:

1) Since LNPs are often associated with strong inflammation further investigation are needed to exclude major tissue damages. The parameters of liver function and body (lung) weight (fig 2g) are encouraging, nonetheless a histopathology evaluation to check for inflammation or injury is of crucial importance.

2) The authors demonstrate CFTR recovery in treated mice by performing swelling assays in mice derived intestinal organoids (Fig 4). Even though intestinal organoids are very useful and widely used experimental models to evaluate CFTR activity nonetheless they amplify the levels of CFTR recovery. This because even a low percentage of repaired cells are sufficient to produce organoid swelling. This is visible in this study by the lack of correlation between CFTR genetic correction (around 2%-fig 4h right panel) and organoid swelling (more than 20% fig 4h left panel). A more accurate evaluation also in consideration that lung should be the primary target and not the intestinal epithelium can be performed by ALI culture of bronchial cells. The technology is mastered by the authors as they perform the transwell experiments in Fig. 5

3) The editing analyses are performed by Sanger seq (TIDER). A more accurate deep seq is highly suggested to increase sensitivity. Deep seq may also solve the problem of high background reported in Fig. 4d (NC sample). Same results should be reported with statistical analyses

Minor comments:

1) Page 5 line 10: "TA clone" is not spelled out in the manuscript. Suggest to move to method and describe in main text as cloning

2) The discussion paragraph is very limited. I suggest extending the section to discuss advantages and disadvantages of LNPs compared to delivery tools so far developed for lung genome editing (including recent Cas9-peptide delivery and other LNPs developed for lungs)

Reviewer #1:

This manuscript describes an exciting study of tissue-specific targeting LNPs for gene editing of Cystic Fibrosis (CF). Wei and colleagues apply their previously developed SORT LNP technology to deliver CRISPR/Cas9-based editing agents to correct CF mutations in cell culture and in vivo. While gene editing efficiency in the lungs of G542X mice was somewhat low (~4%) and variable following systemic delivery of lung SORT LNPs, this study serves as a proof of concept for successful intravenous lung delivery of editing agents using LNPs. Additionally, the demonstration of gene editing of CF stop codon mutations is timely, given that patients with these mutations are not eligible for TRIKAFTA and are in need of other treatment options.

Thank you very much for the encouraging comments. We share your enthusiasm and emphasize that it is important for the broader scientific community to continue to explore all possible approaches to collectively contribute to future advancement of new therapies for CF patients who carry nonsense mutations.

Major Criticisms:

- 1. The authors indicate basal stem cells in the lung as the ideal target for editing therapeutics due to the potential for long-lasting expression of CFTR in epithelial cells, and demonstrate that SORT LNPs are able to transfect these stem cells. However, the true test of the above mentioned hypothesis would be an assessment of the longevity of the editing response or CFTR expression in vivo, which is not provided.*

Discussion: Thank you very much for your suggestion. In this manuscript, we report that reengineered 2nd generation SORT LNPs can reach lung basal cells following intravenous (IV) administration, which is an important step towards durable genome editing therapies. Furthermore, we demonstrate successful CRISPR/Cas homology-directed repair (HDR) genome correction of CFTR in the “gold standard” human bronchial epithelial (HBE) cell model and in genetically engineered CF mice harboring the G542X nonsense mutation. As indicated in the responses below, we have performed additional experiments to further support these significant advancements.

While investigating the long-term editing of CFTR gene (e.g., one year or longer) would provide strong evidence regarding the potential positive consequences of basal cell editing, certain limitations need to be considered. The G542X CF mouse model used in our study does contain the G542X mutations within the genome; however, it does not exhibit pronounced pathological features of CF in the lungs. This makes it less suitable for evaluating long-term changes in pathological phenotypes. Additionally, undertaking an extensive, long-term assessment of editing outcomes in newly-developed models requires a considerably much longer timeframe than the time required for this revision. Often times, one must strike a balance between the time-consuming optimization of early-stage technologies and the goal of promptly sharing their research with the scientific community. This manuscript serves as a proof-of-concept study, demonstrating the effectiveness of our system in repairing multiple CFTR mutations and restoring its function. We anticipate that future optimization of this proof-of-concept technology will lead to a second comprehensive manuscript, which will involve alternative CF models that are better suited to addressing the goal of assessing long-term changes in CF pathology.

- 2. Other recent work, with an alternate nanoparticle delivery system (ref. 40), also provided evidence of delivery to basal cells, which might be a good point of comparison.*

Discussion: We appreciate this feedback, and this very important reference was indeed cited in the original submission. To the best of our understanding, a key difference is that delivery to basal cells using the referenced

PLGA polymer-based peptide nucleic acid (PNA) nanoparticle system was determined by quantifying Cy5 dye-conjugated PLGA nanoparticles, which is a measurement of biodistribution and not necessarily functional delivery. In this current manuscript, we examined Lung SORT LNP delivery of Cre recombinase mRNA in the lox-stop-lox tdTomato mice, whereby only after the more complete process of functional delivery (biodistribution, cellular uptake, endosomal escape, translation of mRNA to protein, nuclear localization of Cre, and DNA editing) can the cells become tdTomato positive. We have herein quantified the percentage of tdTomato-expressing cells within the basal cell population, which accurately reflects the more relevant and significant process of functional delivery.

Action: We have revised the **Main Text** (highlighted in yellow) to bring more specific attention to this reference in multiple instances. Although the carrier type, nucleic acid type, and readout all differ between the referenced work and our current manuscript, we believe it is best to emphasize the importance of this earlier work and not dwell on potential limitations of the previous study as a basis for comparison. As such, we referenced and commented on this important work (ref. 40) in a manner that allows interested readers to locate and read the paper too.

3. *The intestinal organoid studies are interesting and are helpful for testing of the editing agents in tissues from the G542X mouse model but they do not provide evidence of in vivo delivery effectiveness. This seems particularly important in this case, since the in vivo delivered LNPs are designed to target the airways and not the intestines.*

Discussion: Thank you for this comment. Indeed, the *ex vivo* intestinal organoid model served as an effective model for evaluating the intracellular efficacy of editing agents, which is a critical step in the overall delivery process. While the results are indeed less relevant to lung editing, as a proof-of-concept study, the data in intestinal organoids carrying G542X nonsense mutation underscores a substantial restoration of CFTR protein function through DOTAP LNP-mediated editing. This association provides valuable insights into the efficacy of (1) the CRISPR/Cas9 HDR editing system and (2) the delivery system as dual goals of the experiment. To overcome this limitation, we also utilized primary patient-derived HBE model to evaluate the DOTAP LNP-mediated editing and CFTR recovery in human lung cells. We do appreciate your highlighting of this study, as the outcome in some ways highlights capacity of DOTAP-LNPs for multi-cargo delivery to multiple cell types and utility in diverse cellular contexts.

Action: We have edited the **Main Text** to describe both the limitations of the organoid data in terms of relevant to lungs, as well as the potential broader impacts of the data for the ability to transfect non-lung cell types.

4. *An in vivo assessment of phenotypic correction in the airways would be more compelling to show a correlation with genotypic correction. Others have used nasal potential difference measurements in CF mouse models. An in vivo assay such as this should be included.*

Discussion: Thank you for the suggestion! We agree that measurements such as nasal potential difference (NPD) can be a useful correlate to genotypic correction. However, to observe a significant difference in NPD, nanoparticle systems must effectively reach the nasal epithelium. Prior studies correlating correction with function have suggested that 6-25% genomic correction may be required to observe an obvious difference in NPD.^{1, 2} In this study, we optimized our LNP formulation for enhanced lung delivery through intravenous administration, not nasal epithelium delivery. Nevertheless, we do agree that assessment of the nasal epithelium can be a useful correlate and therefore performed a new functional biodistribution assessment focusing on mouse airway compartments (lung, trachea, and nasal epithelium). We found that DOTAP40-LNPs can reach the nasal epithelium, but at approximately 1000-fold lower efficiency compared to lung delivery (**Supplementary Fig. 5**). This discovery is indeed promising, demonstrating successful functional delivery to the nasal epithelium using single-cargo DOTAP40 LNPs. However, with such moderate level of delivery efficiency, we think it is less likely to be able to achieve certain level of gene correction in nasal epithelium using our lung-targeting LNPs for multi-cargo delivery.

Action: We conducted a functional biodistribution assessment and incorporated the results as a part of the new **Supplementary Fig. 5** in the revised SI (highlighted in yellow). This addition highlights the moderate level of protein expression observed in the nasal epithelium of DOTAP40-Luc treated C57BL/6 mice using *ex vivo* imaging.

Supplementary Fig. 5 | DOTAP40 LNPs enabled functional mRNA delivery to airway tissues (lung, trachea, and nasal epithelium). C57BL/6 mice were treated with DOTAP40 LNPs (Luc mRNA, 0.1 mg kg⁻¹, i.v., 20/1 (total lipid/total RNA)). *Ex vivo* airway tissue imaging (a) and quantification of luciferase expression (b) was performed 6 hours after the intravenous injection.

5. *Basolateral as opposed to apical administration of LNPs of ALI cultures would be a better recapitulation of IV delivery.*

Discussion: Thank you for this suggestion. It does make sense that basolateral administration of LNPs to ALI cultures could be a better recapitulation of *in vivo* systemic delivery to the lungs since LNPs reach the differentiated epithelium through systemic IV administration (blood side). Overall, we believe that both methods (apical and basolateral delivery) have value in assessing LNP delivery to HBE cells, as both methods can reveal genome editing and functional outcomes that test the editor (CRISPR/Cas) and delivery vehicle (Lung SORT LNPs) simultaneously. To address this helpful suggestion on basolateral delivery, we performed a new experiment to quantify delivery by this alternative administration route.

Action: To evaluate feasibility of Lung SORT LNPs to transfect ALI cultures by basolateral delivery, we sought a method that could quantify LNP delivery by this alternative administration route. To that end, we encapsulated tdTomato mRNA using DOTAP10 LNP (DOTAP10-tdTOM) and delivered them to fully differentiated HBE cells from either the apical side or the basolateral side. This allowed quantification of functional delivery (cellular uptake, endosomal escape, translation of mRNA to protein) in a way that allowed quantification by flow cytometry. As shown in newly added **Supplementary Fig. 15** in the revised SI (highlighted in yellow), DOTAP10-tdTOM (tdTomato mRNA, 12ug per well) transfected >20% of basal cells from the basolateral side. We appreciate the suggestion, as this data strengthens the conclusion that our LNPs can transfect basal cells following systemic administration.

Supplementary Fig. 15 | Apical and basolateral delivery of DOTAP10-tdTOM in P3 fully differentiated HBE culture. a, Representative flow cytometry images (a) and quantitative analysis (b) of tdTOM fluorescent protein expressing (TdTOM⁺) cells in basal cells and differentiated cell populations. HBE cells were collected 24 hours after a single treatment of DOTAP10-tdTOM (12ug per well) from either apical side or basolateral side. Untreated HBE cells were used as control. c, Flow cytometry gating strategy for HBE basal cells. Single cells collected from each transwell were gated. Alive cells were gated using SYTOX Blue dye. HBE basal cells (NGFR-PE-Cy7 positive) expressing tdTOM fluorescence (tdTOM positive) were analyzed by flow cytometry. Data are shown as mean±s.e.m. (n=2).

6. *There is no mention of off-target effects of Cas9 editing in terms of editing in off-target organs or off-target genomic sites in the lungs. This analysis should be done as a standard assessment of safety and genotoxicity of CRISPR/Cas9-based editing.*

Discussion: Thank you for the valuable suggestion! We agree with the reviewer that it is important to evaluate off-target effects in terms of editing in off-target organs or off-target genomic sites in the lungs, so we have supplemented the experiments and added corresponding data in the revised SI as **Supplementary Fig. 10** and **Supplementary Fig. 12**.

Action: We initially employed a prediction tool (CRISPOR³, <http://crispor.tefor.net/>) to identify the top-six off-target sites of sgG542X. Subsequently, we conducted amplification of these potential off-target sites from DOTAP40-HDR treated G542X mouse lungs and quantified off-target editing using NGS deep sequencing. Untreated G542X mouse lungs were used as control (NC). The result demonstrated minimal to no off-target editing, with rates below 0.06%, at the top-ranked off-target sites. This result has been added into the revised SI as **Supplementary Fig. 12**.

a

Target	Gene	gRNA Sequence	PAM	MIT-Score	Off-target editing efficiency (%)
On Target	CFTR	GACAACACAGTTCTCTGAGA	AGG		
OT1	intergenic:Mir568	GCCACACAGTGCCTCTGAGA	GGG	1.38	0.04
OT2	intergenic:Plxnc1-Gm29684	AGTAACACATTTCTCTGAGA	AGG	1.30	0.00
OT3	intron:Gxylt2	GACAACAACACTCTCTGAGA	GGG	1.10	0.04
OT4	intron:Xrcc4	AATAAGAGAGTTCTCTGAGA	TGG	0.81	0.00
OT5	intergenic:Slc6a15-Gm8764/Gm6763/Gm4340/Gm21312/Gm21304/Gm21293/Gm20765	AGCATAACAGTTCTCTGAGA	GGG	0.79	0.04
OT6	intergenic:Syn2-Pparg	CACTGACAGTTCTCTGAGA	AGG	0.79	0.06

b

Supplementary Fig. 12 | No off-target editing was detected after treatment with DOTAP40-HDR LNPs. A. Sequence information of top-six predicted off-target sites of sgG542X was listed in the table. B, The top 6 potential off-target sites were amplified by PCR and analyzed using NGS deep sequencing. NP only treatment group was used as negative control (NC).

To assess off-organ targeting, we performed a functional biodistribution study by delivering luciferase mRNA using DOTAP40 LNPs. C57BL/6 mice were dissected 6 hours after a single injection of DOTAP40-Luc (luciferase mRNA, 0.1 mg kg⁻¹, IV). The luciferase expression in major organs (heart, lung, liver, spleen, and kidney) was measured by *ex vivo* imaging and the fold changes in luciferase expression were calculated by comparing to untreated C57BL/6 mice. We found luciferase expression was predominantly observed in the lungs compared to the other organs tested.

Additionally, we used Ai14 reporter mice to access the potential off-target Cas9 editing in off-target organs. tdTOM mice were sacrificed one week after the third weekly treatments of DOTAP40-NHEJ (2 mg kg⁻¹, Cas9 mRNA/sgTOM1=2:1, wt/wt) using the same dosing regimen as employed in the *in vivo* correction experiment in G542X mice. Tissues including brain, heart, lung, liver, spleen, kidney, pancreas, GI tract, and reproductive organs (Ovaries/Testes) were collected to evaluate potential the off-target organ editing. The fold changes in fluorescence signals in different organs were measured by *ex vivo* imaging. The result showed that Cas9-mediated *in vivo* editing was predominantly observed in the lung. These results have been added into the revised SI as **Supplementary Fig. 10.**

Supplementary Fig. 10 | Lung SORT DOTAP40 LNP facilitates functional mRNA and gene editor delivery predominantly in mouse lungs, with minimal off-organ delivery observed. a, Biodistribution of functional mRNA delivery after DOTAP40-Luc treatment. C57BL/6 mice were intravenously administrated with DOTAP40-Luc formulation (0.1 mg kg⁻¹ Luc mRNA, 20/1 (total lipid/total RNA)) once a week, three weeks in total. Mice were

sacrificed 6 hours after the last injection and dissected to collect major organs (heart, lung, liver, spleen, and kidney). Biodistribution of functional luciferase mRNA was measured by *ex vivo* imaging using AMI imaging systems. Untreated C57BL/6 mice were used as control. Data are reported as fold changes in average radiance compared to untreated C57BL/6 mice. b, *In vivo* off-organ editing assessment after DOTAP40-NHEJ treatment. Ai14 tdTOM reporter mice were intravenously administrated with DOTAP40-NHEJ formulation (2 mg kg⁻¹ Cas9 mRNA:sgTOM1=2:1, wt/wt, 20/1 (total lipid/total RNA)) once a week and three weeks in total. Mice were sacrificed one week after the last injection and dissected to collect organs (brain, heart, lung, liver, pancreas, spleen, kidney, GI tract and reproductive organs). Off-organ editing was assessed by measuring tdTOM fluorescence production after successful *in vivo* editing in reporter mice using AMI imaging systems. Untreated Ai14 mice were used as control. Data are reported as fold changes in average radiance compared to untreated Ai14 mouse. All data are shown as mean±s.e.m. (n=3 biologically independent samples).

7. Immunogenicity of LNPs (ex. serum and BALF cytokine levels) should be assessed in addition to the toxicity indicators assessed.

Discussion: Thank you for the suggestion. In addition to assessing liver and kidney function (**Figure 2g**) and monitoring body weight (**Figure 2h, 2i**) after LNP treatment, we also performed H&E staining of mouse tissues (heart, liver, lung, kidney, and spleen) for histopathology evaluation (**Supplementary Fig. 3**). No significant *in vivo* toxicity was observed in DOTAP40 LNP group compared to DDAB30 LNP and lipopolysaccharide (LPS) groups. We recognized that the immunogenicity of LNPs could be a concern whereby LNPs could potentially lose efficacy upon repeat administration. We then performed additional experiments to evaluate these potential immunogenicity factors.

Action: We prepared DOTAP40 LNPs and evaluated the delivery efficiency after repeated administration of DOTAP40 LNPs. The result showed that DOTAP40 LNPs maintained similarly high delivery efficacy, indicating that immune clearance of DOTAP40 LNP did not occur after repeated administration. This result has been added into the revised SI as **Supplementary Fig. 4**.

Supplementary Fig. 4 | DOTAP40 LNPs allowed repeat dosing without loss of efficacy. a, C57BL/6 mice were treated with DOTAP40-Luc (Luc mRNA, 0.1 mg kg⁻¹, IV) three times in total, one week apart. Whole body *in vivo* imaging (b) and quantification of luciferase expression (c) was performed 6 hours after each injection. *Ex vivo* major organ imaging (b) was performed 6 hours after the third injection. Data are shown as mean±s.e.m. (n=3 biologically independent animals). Two-tailed unpaired t-tests were used to determine the significance of the comparisons of data (*P<0.05; **P<0.01; ***P<0.001).

8. Some of the claims in the manuscript are vague, and the limitations of the study would be clearer if the authors could be more specific. For example, the claim that basal cells are edited is based on studies with Cre mRNA in a reporter mouse, not CFTR editing in a CF mouse. Successful delivery to basal cells in one model does not necessarily translate to another gene target and animal model.

Discussion: Thank you for the comment! We have revised the description of the results in the Main Text, to make them more accurate. We also expanded the Discussion section at the end of the manuscript, as recommended.

Minor Criticisms:

1. Do lung SORT LNPs reach other airway cells (ex. trachea, nose)? A biodistribution assessment of these tissues would be helpful.

Discussion: Thank you for your question. As suggested, a functional biodistribution assessment was performed and included as a part of new **Supplementary Fig. 5** in the revised SI. We observed moderate level of protein expression in trachea and nasal epithelium of DOTAP40-Luc treated C57BL/6 mice using *ex vivo* imaging, but much less than that observed in the lung.

2. Why is there a positive HDR editing signal in NC control samples in Figure 4? Shouldn't editing readouts for these controls be closer to 0%?

Discussion: The original HDR editing efficiency was performed by Sanger sequencing and analyzed using TIDER analysis. Since TIDER is an online tool to calculate gene editing efficiency, high noise may be observed when the editing efficiency of the treatment group is relatively low. That may be the reason why there is a positive HDR editing signal in NC control samples in **Figure 4**. To make it more accurate, we further analyzed the HDR editing efficiency of DOTAP40-HDR treatment treated mouse lungs using NGS sequencing.

Action: To address the problem mentioned by the reviewer, we performed deep sequencing and observed up to 2.34% correction in DOTAP40-HDR treated mouse lungs (with a mean 1.24% of correction), whereas minimal to no correction was detected in NC control samples based on analysis of NGS data (**Figure 4d**). There is a significant difference between NC control and DOTAP40-HDR groups with a P value equal to 0.0168, unpaired t-test. To make it more accurate, we have updated **Figure 4d** with more accurate NGS data in the revised manuscript.

Figure. 4d | HDR efficiency in DOTAP40-HDR treated mouse lung by NGS and CRISPResso2 analysis.

3. Are HeLa cells the best screening cell line to use for *in vitro* studies if the intended cell type is primary HBEs grown at different culture conditions?

Discussion: Thank you for the question. The sequence of sgRNA and HDR template, as well as the internal ratio of Cas9 mRNA/sgRNA/HDR during encapsulation, play a crucial role in the overall gene correction efficiency of DOTAP LNPs. Therefore, it is necessary to optimize these parameters before delivering them to G542X mice for treatment. To accelerate the optimization process, we established HeLa cells carrying the same G542X mutation as G542X mice (referred to as HeLa-G542X). We used the HeLa-G542X cell model for optimizing the sequence and packaging method since these cells are highly abundant and passagable. This allowed us to determine the final sequence and composition ratios that were subsequently used for evaluating the editing efficiency in G542X mice, which are scarce. As for the F508del mutation, we had primary undifferentiated HBE cells carrying homozygous F508del CF mutations, which we directly used for optimizing the F508del HDR template (**Supplementary Fig. 13**).

4. *It would be helpful to note changes, if any, in LNP characteristics (size, surface charge, etc.) and biodistribution for the various formulations incorporating gene editing cargo (mRNA, sgRNA, ssDNA) during optimization of these LNPs.*

Discussion: In our previous manuscripts, we have systematically evaluated all of these parameters, and determined that tissue tropism tightly correlates with the chemistry and amount of the SORT molecule added.^{4,5} We have also evaluated properties including size, surface charge, and cargo type in prior manuscripts.⁶⁻⁹ Through these investigations, we determined that SORT biodistribution / tropism most closely correlates with the chemistry and amount of the SORT molecule. It does not correlate with size, surface charge, or cargo type.

Action: We agree that cataloging characterization data for the SORT LNPs tested in this manuscript would provide readers with more information and a reference point to understand the work. As suggested, we added a new **Supplementary Table 1** to the **SI** to detail relevant data on the LNP formulations tested including size, PDI, zeta potential, and mRNA encapsulation efficiency.

5. *Why was a weight ratio used for Cas9 mRNA/sgRNA LNP optimization as opposed to molar ratios? Providing conversions to molar ratios might be helpful for many readers looking at similar approaches. In addition, it would be interesting to know the weight ratio of total nucleic acid:lipid in the final preparations (I believe 40:1 or 20:1 from Fig 4), and estimates of how many copies of each molecule are in each LNP particle.*

Discussion: Thanks for all the comments! 1) We optimized the LNP-HDR formulation by altering the weight ratio of Cas9 mRNA, sgRNA, and HDR template in the manuscript, to make the formulation preparation much easier. To make sure the audience to clearly know the exact molar ratio among these three components, we also provide their molar ratios (**Supplementary Fig. 7c**); 2) The weight ratio of total lipid/total nucleic acid in the final preparations was 20:1 for *in vivo* studies. We have emphasized the information in the revised manuscript (**highlighted in yellow**); 3) Quantifying the exact number of respective molecules per LNP, especially when it contains multiple payloads including mRNA, sgRNA, and single-stranded HDR DNA, is a highly complex task. Unfortunately, there are currently no methods available that can precisely quantify these molecules in LNPs due to their incompatibility with established quantification methods.

Action: As suggested, we have added a new table that includes both the weight ratios and molar ratios of mRNA/sgRNA/HDR in the revised **SI** (**Supplementary Fig. 7c**). The description on the weight ratio of total lipid/total nucleic acid in the final preparations for *in vivo* studies has been added to the revised manuscript.

Cas9 mRNA/sgRNA/HDR (weight ratio)	Molar Ratio		
	Cas9 mRNA	sgRNA	HDR
1:1:0.25	1	45	10
1:1:0.5	1	45	19
1:1:1	1	45	38
1:1:3	1	45	114
1:1:6	1	45	229
1:1:10	1	45	382
2:1:6	1	23	114
0.5:1:6	1	91	458
0.25:1:6	1	181	916
0.1:1:6	1	453	2289

Supplementary Fig. 7c | The weight ratios and molar ratios of Cas9 mRNA/sgRNA/HDR used in the DOTAP10 LNP formulations.

6. How were HDR templates designed/chosen? This is unclear.

Discussion: Single stranded DNA (ssDNA) was selected as the format of HDR templates in this study due to its compact nature and lower toxicity compared to plasmid-based donor vectors and double stranded DNA (dsDNA) donors, which could improve knock-in efficiency.¹⁰ Additionally, ssDNA can be synthesized on a solid support allowing integration of phosphorothioate modifications on both end to enhance its stability in cells and *in vivo*, thus improve the editing efficiency.¹¹ The iterative course of our template design involved initial construction of three distinct HDR templates for each target, each characterized by the incorporation of the intended edit, positioned between homology arms of varying lengths and template polarity (i.e. template in sense or antisense orientation). The optimal HDR template, specific to the F508del mutation, was identified through a correction assessment employing primary undifferentiated CF HBE cells with homozygous F508del mutation as illustrated in **Supplementary Fig. 13**. Moreover, for the G542X mutation, a similar approach was undertaken. The optimal HDR template was developed and assessed within reporter cells harboring the mouse G542X mutation. This exercise afforded us valuable insights into the performance characteristics of the LNP-HDR formulation specific to the F508del or G542X context respectively.

Action: We added a new figure to better describe this workflow and sequences as **Supplementary Fig. 13** in the revised SI.

Supplementary Fig. 13 | HDR template optimization using P1 undifferentiated HBE culture with *CFTR*^{F508del/F508del} mutation. a, Patient derived undifferentiated HBE cells with F508del mutation were treated with DOTAP10-HDR LNP for three days before cell collection for sequence analysis. b, A list of HDR templates tested for HDR template optimization. b, HDR correction level analyzed by Sanger sequencing and TIDER analysis.

Reviewer #2:

Discussion: Thank you for your help and constructive feedback!

Reviewer #3:

SIGNIFICANCE:

The development of CFTR modulator compounds has provided a large benefit for roughly 90% of CF patients. However, the remaining 10% do not respond to these modulators and require other therapeutic approaches. While gene correction via CRISPR/Cas would be a significant advance towards such therapies, this approach is currently limited by the lack of efficient genome editor delivery carriers. The current study reports development of improved Lung Selective Organ Targeting (SORT) Lipid Nanoparticles (LNPs) that can efficiently deliver Cas9 mRNA, sgRNA, and donor ssDNA templates for precise homology-directed repair (HDR)-mediated gene correction in ex vivo and in vivo CF models. These new LNPs successfully corrected the G542X CFTR mutation in G542X/G542X mice and in patient-derived F508del/F508del human bronchial epithelial (HBE) cells, leading to restored expression of CFTR protein and chloride transport. This more efficient formulation of LNPs and the demonstration of its utility to promote gene correction is an important development in the CF field.

I think this is a solid manuscript describing the development and utility of a new formulation of LNP for the delivery of gene-editing components.

Discussion: Thank you for acknowledging the significant challenges faced by cystic fibrosis patients, who currently lack effective treatments, and for your positive feedback on our manuscript.

I have the following questions & comments:

1. *Page 5, top paragraph (and Figure 4c): You chose to do 3 weekly injections of your cocktail and then harvest after one additional week. Did you try any other dosing schedules (shorter or longer)? If not, how do you know this is optimal?*

Discussion: Thank you for this question. During our investigation, we examined both single and double dosing regimen (one week apart); however, these approaches yielded lower (nearly undetectable) gene correction in whole lungs, as analyzed by Sanger sequencing, compared to the selected regimen of three weekly doses. We believe there is an additive effect on gene correction with multiple injections. While a more comprehensive exploration into dosing regimens and the frequency of administration would provide nuanced insights, it would also require more genetically engineered G542X CF mice. Unfortunately, we currently do not have enough available mice and time to conduct such a study, but we believe higher gene correction efficiency may be achieved if we further increase the dosing regimens.

2. *Page 5, top paragraph (and Figure 4d): You state in the text that you obtained a correction rate for G542X in vivo of 4%. However, figure 4d seems to indicate a rate of 3%. Please address this disparity.*

Discussion: Thank you for catching this point. The *in vivo* correction rate of G542X mouse lung is mean 2.9% (up to 5.1% in one mouse) by Sanger sequencing and TIDER analysis. Here, for this revision, we performed deep sequencing for a more precise assessment. The CRISPResso2¹² analysis results showed that up to 2.34% correction in DOTAP40-HDR treated mouse lungs (with a mean 1.24% of correction), whereas minimal to no correction was detected in NC control samples based on analysis of NGS data. To make it more accurate, we have updated **Figure 4d** with more accurate NGS data in the revised manuscript. We have also performed statistical analysis, revealing p value of 0.0168.

Figure. 4d | HDR efficiency in DOTAP40-HDR treated mouse lung by NGS and CRISPResso2 analysis.

3. You suggest that by isolating all lung cells, the rate of editing is likely diluted through isolation of cells assumed to be “irrelevant for CF treatment”. If you sorted lung cells using an epithelial cell marker, could you get a better estimate of the efficiency of correction?

Discussion: Thanks for your valuable suggestion! In fact, the lungs are composed of numerous different cell types, and only some of them express CFTR proteins. Therefore, we agree that evaluating gene correction efficiency from bulk lung tissues could potentially dilute the actual editing efficiency significantly. In contrast, sorting lung epithelial cells may yield more accurate correction efficiency.

Action: Because of the time limitation and available mice numbers, we were unable to try this method after gene correction treatment of DOTAP40-HDR in G542X mice. However, to address this point, we here provide additional data in **Figure 3g** and **3h**. We examined Lung SORT LNP delivery of Cas9 mRNA/sgTOM1 with or without HDR template in the lox-stop-lox tdTomato reporter mice, whereby only after functional delivery of Cas9 mRNA and sgRNA to achieve double or triple deletion of stop cassettes can the cells become tdTomato positive (tdTOM⁺). We have herein quantified the percentage of tdTomato-expressing cells within the bulk lung cells, CD31 expressing Lung endothelial cells, CD45 expressing lung immune cells, and EpCam expressing lung epithelial cells (**Figure 3g, 3h**). The comprehensive flow cytometry analysis brought significant insights. Within the bulk lung tissue, an editing rate of 12.7% (DOTAP40-NHEJ) and 12.4% (DOTAP40-HDR) within bulk lung was observed. Notably, this rate exhibited a marked augmentation by nearly 1.5-fold, reaching 18.8% (DOTAP40-NHEJ) and 1.8-fold, reaching 21.7% (DOTAP40-HDR), within the subset of lung epithelial cells from the bulk lung tissue. Statistical analyses have been performed and added to **Figure 3g** and **3h** in the revised **Main Text**. There was a significant difference in tdTomato positivity between bulk lung and lung epithelium cells population with a P value equal to 0.0011 (DOTAP40-NHEJ) and 0.0218 (DOTAP40-HDR). We believe that the data comparisons confirm a significant augment in editing within lung epithelial cell population compared to bulk lung cells. The inclusion of this proof-of-concept cell sorting strategy constitutes a noteworthy advancement, as it offers a more refined approach for estimating the efficiency of *in vivo* correction within the subset of CF-relevant cellular populations.

Figure. 3f, 3g, 3h | *In vivo* gene editing in Ai14 mouse lungs. a, Ai14 tdTOM reporter mice were used to evaluate CRISPR/Cas gene editing in bulk lung and the subset of CF-relevant cellular populations. Ai14 tdTOM reporter mice were treated once a week with (b) DOTAP40-NHEJ (Cas9 mRNA:sgTOM1=2:1, wt/wt) and (c) DOTAP40-HDR(Cas9 mRNA: sgTOM1: ssDNA HDR template for CF G542X=2:1:3, wt/wt/wt) formulations respectively with a total RNA dosage of 1 mg kg⁻¹, IV. One week following the third injection, mice were sacrificed, and the lungs were collected for flow cytometry analysis to determine the proportion of tdTOM⁺ cells in each different cell type. Data are shown as mean±s.e.m. (n=3 biologically independent animals). Two-tailed unpaired t-tests were used to determine the significance of the comparisons of data (*P<0.05; **P<0.01; ***P<0.001).

4. Page 5, line 22: you state that >20% of organoids swelled after treatment, while your HDR correction rate in intestinal organoids was only 2.5% (Figure 4i). Is this because each organoid is made up of a cluster of starting cells? Please explain.

Discussion: Thank you for your comments. *Ex vivo* intestinal organoid is a valuable model for assessing the CFTR function recovery ability of LNP-HDR formulation after correcting G542X mutation in the genome. However, it is a very sensitive assay that can potentially magnify the extent of CFTR functional restoration. Others have observed that moderate efficacy can lead to organoid swelling.¹³ This phenomenon also occurred in our study, where we observed a disparity between the correction of CFTR gene (~2.5%, **Figure 4i**) and the subsequent organoid swelling (~20%, **Figure 4h**). To enhance the precision of our functional restoration evaluation, we also utilized primary HBE model using air-liquid interface (ALI) culture of bronchial cells derived from a CF patient (**Figure 5**) to evaluate the gene correction efficacy and function recovery of CFTR proteins of our LNP-HDR formulations.

Action: We have added more description in the revised **Main Text** (highlight in yellow) to clarify the relationship between gene correction and functional restoration in *ex vivo* intestinal organoids.

5. Figure 5e: What level of correction does this AUC represent? In other words, please indicate the AUC obtained with WT HBE cells for comparison.

Discussion: Thank you very much for this suggestion! We fully agree with this excellent suggestion. We performed this experiment, and added the data of WT HBE cells in the **Figure 5d** and **5e** of revised manuscript for comparison.

Action: As suggested, we measured the chloride conduction current of wild-type (WT) CFTR from HBE cells obtained from a healthy human donor (without CF) using the same culturing condition and measurement method. Comparing to WT CFTR presumed to have 100% CFTR activity (AUC 21.6 μA/cm²*min), DOTAP10-HDR alone effectively achieved full restoration of the CFTR function (AUC 22.5 μA/cm²*min) (**Figure 5d, 5e**). We have added this promising result and description in the revised **Main Text** to clarify the level of function restoration represented by the AUC data.

Figure. 5d, 5e | Full CFTR function restoration from HBE culture expressing mutant CFTR^{F508del/F508del} was observed after DOTAP10-HDR treatment, compared to wild-type (WT) CFTR activity.

6. *Figure 5h: It is not clear that some of these data are significantly different. In particular, please provide p values comparing DOTAP vs. DOTAP + VX-809. Is there a significant additive effect?*

Discussion: Thank you very much for this suggestion. As suggested, we have analyzed the significance between different groups and provide p values in **Figure 5h** of the revised manuscript.

Action: Statistical analyses have been performed and added to **Figure 5h**. Two-tailed unpaired t-tests were used for comparison. There was a significant difference between DMSO and DOTAP10-HDR groups with a P value < 0.0001; between DOTAP10-HDR and DOTAP10-HDR + VX809 groups with a P value equal to 0.0002; between VX-809 and DOTAP10-HDR + VX-809 groups with a P value < 0.0001. We believe that the data comparisons confirm a significant additive effect.

Figure. 5h | CFTR function measurement using HBE culture expressing mutant CFTR^{F508del/F508del} after DOTAP10-HDR treatment at undifferentiated stage with or without the addition of VX-809.

7. *Figure 5n: It is not clear that DMSO and DOTAP are significantly different, or if VX-809 and DOTAP + VX-809 are significantly different. In addition, if DMSO and DOTAP are the same, then the further increase from VX-809 vs. DOTAP + VX-809 could be an artifact, possibly due to greater cell permeabilization due to the lipids, thus allowing more VX-809 to enter.*

Discussion: Thank you very much for this suggestion.

Action: Statistical analyses have been performed and added to **Figure 5n** in the revised manuscript. Two-tailed unpaired t-tests were used for comparison between the respective two groups. There was a significant difference between DMSO and DOTAP10-HDR groups with a P value equal to 0.042.

Figure. 5n | CFTR function measurement using HBE culture expressing mutant CFTR^{F508del/F508del} after DOTAP10-HDR treatment at differentiated stage with or without the addition of VX-809.

Additionally, we added new data on transepithelial resistance (**Supplementary Fig. 14**) in the revised SI, which is indeed an established metric for cell health and the integrity of the epithelial layer. No significant difference was observed between DMSO and DOTAP10-HDR groups by performing a two-tailed unpaired t-test, indicating no change to epithelial integrity following LNP treatment. We believe that this set of transepithelial resistance data confirms the maintenance of HBE cell integrity on porous support after DOTAP10-HDR treatment.

Supplementary Fig. 14 | Transepithelial resistance using HBE culture expressing mutant CFTR^{F508del/F508del} after DOTAP10-HDR treatment at differentiated stage with or without the addition of VX-809.

Reviewer #4:

The manuscript by Tuo Wei from Daniel Siegwart group reports on lipid nanoparticles optimized from a former study that are optimized for mRNA delivery of Cas9 together with the sgRNA and a donor ssDNA to perform HDR in lungs. The authors analyze the efficacy of LNPs delivery in vitro and in mice to test the tissue tropism and editing efficacy in lungs. CFTR functional recovery is tested in vitro using intestinal organoids derived from treated mice and primary human epithelial cells from patients (F508del).

This study stems from a previous work by the same group developing LNP and here they further optimize LNPs for *in vivo* delivery and HDR editing. The reported method of delivery has obvious implications for the advancement therapeutic strategies for cystic fibrosis.

The work is very well performed and includes convincing *in vivo* data.

Discussion: Thank you for these kind comments.

Major comments:

1. Since LNPs are often associated with strong inflammation further investigation are needed to exclude major tissue damages. The parameters of liver function and body (lung) weight (fig 2g) are encouraging, nonetheless a histopathology evaluation to check for inflammation or injury is of crucial importance.

Discussion: Thank you for this valid comment. In addition to assessing liver and kidney function and monitoring body weight (**Figure 2g**) after LNP treatment, we also performed H&E staining of mouse tissues (heart, liver, lung, kidney and spleen) for histopathology evaluation (**Supplementary Fig. 3**). No significant *in vivo* toxicity was observed in DOTAP40 LNP treatment group compared to DDAB30 LNP and lipopolysaccharide (LPS) groups.

Supplementary Fig. 3 | Representative H&E staining images of mouse tissues after treatment with PBS, DDAB30, DOTAP40 and LPS. Mice were treated with DOTAP40 and DDAB30 LNPs (1.5mg kg⁻¹ mCherry mRNA, i.v., 40/1 (total lipid/total RNA)). PBS treatment (i.v.) was used as a negative control, and lipopolysaccharide (LPS, i.p., 5mg kg⁻¹) treatment was used as a positive control. LPS group mice were sacrificed at day 2 after treatment and other groups were killed at day 5. At sacrifice time point, all the organs (heart, liver, spleen, lung and kidney) were collected and sectioned for H&E staining. Obvious tissue damages in heart, liver and lungs were observed when treated with DDAB30 LNPs and LPS, while DOTAP40 LNPs treatment showed low toxicity. Scale bar: 100 μ m.

2. The authors demonstrate CFTR recovery in treated mice by performing swelling assays in mice derived intestinal organoids (Fig 4). Even though intestinal organoids are very useful and widely used experimental models to evaluate CFTR activity nonetheless they amplify the levels of CFTR recovery. This because even a low percentage of repaired cells are sufficient to produce organoid swelling. This is visible in this study by the lack of correlation between CFTR genetic correction (around 2%-fig 4h right panel) and organoid swelling (more than 20% fig 4h left panel). A more accurate evaluation also in consideration that lung should be the primary target and not the intestinal epithelium can be performed by ALI culture of bronchial cells. The technology is mastered by the authors as they perform the transwell experiments in Fig. 5

Discussion: We appreciate this comment. As noted in the response to **Reviewer #3** above, the known caveats of the organoid swelling assay may be overcome by rigorous evidence of editing in various cell lines, including differentiated HBEs, and mouse lungs from G542X mouse. Indeed, we have focused on lung editing in this manuscript. While all G542X mice studies were performed in Cleveland, all ALI HBE studies were performed in Dallas. We are unfortunately not able to keep bronchial cells from G542X mice alive between the cities to conduct the proposed study. We did discuss animal transfer between Case Western Reserve University and UT Southwestern Medical Center, but the required quarantine time exceeds the lifespan of these diseased animals, so that too is not a possibility.

3. *The editing analyses are performed by Sanger seq (TIDER). A more accurate deep seq is highly suggested to increase sensitivity. Deep seq may also solve the problem of high background reported in Fig. 4d (NC sample). Same results should be reported with statistical analyses.*

Discussion: Thank you for these suggestions. As suggested, we performed deep sequencing to evaluate the accurate editing % of the whole mouse lung from mice treated with DOTAP40-HDR. This data has been added into Figure 4d of the revised manuscript by replacing the data analyzed by Sanger sequencing data.

Action: Deep sequencing was performed using NGS. We replaced the data in **Figure 4d** with NGS data analyzed by CRISPResso2. We also performed the statistical analyses using an unpaired t-test. There is a significant difference between NC and DOTAP40-HDR groups with a P value equal to 0.0168.

Figure. 4d | HDR efficiency in DOTAP40-HDR treated mouse lung by NGS and CRISPResso2 analysis.

Minor comments:

1. *Page 5 line 10: “TA clone” is not spelled out in the manuscript. Suggest to move to method and describe in main text as cloning*

Action: Thank you for this suggestion. We have moved “TA clone” to the method section and have described it in the Main Text as cloning.

2. *The discussion paragraph is very limited. I suggest extending the section to discuss advantages and disadvantages of LNPs compared to delivery tools so far developed for lung genome editing (including recent Cas9-peptide delivery and other LNPs developed for lungs)*

Discussion: Thank you for this suggestion.

Action: We have extended more discussion on advantages and disadvantages of LNPs compared to delivery tools so far developed for lung genome editing in the revised manuscript (**highlighted in yellow**).

1. Johnson, L.G. et al. Efficiency of gene transfer for restoration of normal airway epithelial function in cystic fibrosis. *Nat. Genet.* **2**, 21-25 (1992).
2. Zhang, L. et al. CFTR Delivery to 25% of Surface Epithelial Cells Restores Normal Rates of Mucus Transport to Human Cystic Fibrosis Airway Epithelium. *PLoS Biol.* **7**, e1000155 (2009).
3. Concordet, J.-P. & Haeussler, M. CRISPOR: intuitive guide selection for CRISPR/Cas9 genome editing experiments and screens. *Nucleic Acids Res.* **46**, W242-W245 (2018).
4. Cheng, Q. et al. Selective ORgan Targeting (SORT) nanoparticles for tissue specific mRNA delivery and CRISPR/Cas gene editing. *Nat. Nanotechnol.* **15**, 313-320 (2020).
5. Dilliard, S.A., Cheng, Q. & Siegwart, D.J. On the mechanism of tissue-specific mRNA delivery by selective organ targeting nanoparticles. *Proc. Natl. Acad. Sci. USA* **118**, e2109256118 (2021).
6. Wang, X. et al. Preparation of selective organ targeting (SORT) lipid nanoparticles (LNPs) using multiple technical methods for tissue-specific mRNA delivery. *Nat. Protoc.* **18**, 265-291 (2022).
7. Liu, S. et al. Membrane-destabilizing ionizable phospholipids for organ-selective mRNA delivery and CRISPR-Cas gene editing. *Nat. Mater.* **20**, 701-710 (2021).
8. Farbiak, L. et al. All-in-one dendrimer-ased lipid nanoparticles enable precise HDR-mediated gene editing in vivo. *Adv. Mater.* **33**, e2006619 (2021).
9. Zhang, D. et al. Enhancing CRISPR/Cas gene editing through modulating cellular mechanical properties for cancer therapy. *Nat. Nanotechnol.* **17**, 777-787 (2022).
10. Bollen, Y., Post, J., Koo, B.K. & Snippert, H.J.G. How to create state-of-the-art genetic model systems: strategies for optimal CRISPR-mediated genome editing. *Nucleic Acids Res.* **46**, 6435-6454 (2018).
11. Liang, X., Potter, J., Kumar, S., Ravinder, N. & Chesnut, J.D. Enhanced CRISPR/Cas9-mediated precise genome editing by improved design and delivery of gRNA, Cas9 nuclease, and donor DNA. *J. Biotechnol.* **241**, 136-146 (2017).
12. Clement, K. et al. CRISPResso2 provides accurate and rapid genome editing sequence analysis. *Nat. Biotechnol.* **37**, 224-226 (2019).
13. Geurts, M.H. et al. CRISPR-Based Adenine Editors Correct Nonsense Mutations in a Cystic Fibrosis Organoid Biobank. *Cell Stem Cell* **26**, 503-510.e507 (2020).

Reviewers' Comments:

Reviewer #1:

Remarks to the Author:

The authors have been very responsive to the reviewers comments in this revised manuscript. My major concern was that the duration and phenotypic impact of the gene editing was not sufficiently addressed. It is still not addressed, but given the strong quality of the work presented in the original manuscript, and the additional data provided in response to all of the other critiques, I no longer consider this a major concern.

Reviewer #2:

Remarks to the Author:

Reviewer #3:

Remarks to the Author:

This revised manuscript from Siegwart and colleagues study reports development of improved Lung Selective Organ Targeting (SORT) Lipid Nanoparticles (LNPs) that can efficiently deliver Cas9 mRNA, sgRNA, and donor ssDNA templates for precise homology-directed repair (HDR)-mediated gene correction in ex vivo and in vivo CF models. These new LNPs successfully corrected the G542X CFTR mutation in G542X/G542X mice and in patient-derived F508del/F508del human bronchial epithelial (HBE) cells, leading to restored expression of CFTR protein and chloride transport. This more efficient formulation of LNPs and the demonstration of its utility to promote gene correction is an important development in the CF field.

The rebuttal from the authors regarding the first round of reviews was very responsive and thorough. They adequately addressed my concerns and suggestions, and I have no further concerns. Very nice work!

Reviewer #4:

Remarks to the Author:

The authors addressed all issues raised during the revision. This is a solid study giving strong contributions to the gene therapy field for cystic fibrosis. I suggest accepting for publication

Reviewer #1:

The authors have been very responsive to the reviewers comments in this revised manuscript. My major concern was that the duration and phenotypic impact of the gene editing was not sufficiently addressed. It is still not addressed, but given the strong quality of the work presented in the original manuscript, and the additional data provided in response to all of the other critiques, I no longer consider this a major concern.

Thank you very much for the valuable comments! In the future, we hope to have the opportunity to systematically explore the duration and phenotypic impact of the gene editing when better CF mouse models become available. We thank the reviewer again for the critical evaluation of our manuscript and the constructive comments that helped us improve the work.

Reviewer #2:

I co-reviewed this manuscript with one of the reviewers who provided the listed reports. This is part of the Nature Communications initiative to facilitate training in peer review and to provide appropriate recognition for Early Career Researchers who co-review manuscripts..

Thank you for your effort in assessing our manuscript and helping us further improve the quality of our work!

Reviewer #3:

This revised manuscript from Siegwart and colleagues study reports development of improved Lung Selective Organ Targeting (SORT) Lipid Nanoparticles (LNPs) that can efficiently deliver Cas9 mRNA, sgRNA, and donor ssDNA templates for precise homology-directed repair (HDR)-mediated gene correction in ex vivo and in vivo CF models. These new LNPs successfully corrected the G542X CFTR mutation in G542X/G542X mice and in patient-derived F508del/F508del human bronchial epithelial (HBE) cells, leading to restored expression of CFTR protein and chloride transport. This more efficient formulation of LNPs and the demonstration of its utility to promote gene correction is an important development in the CF field.

The rebuttal from the authors regarding the first round of reviews was very responsive and thorough. They adequately addressed my concerns and suggestions, and I have no further concerns. Very nice work!.

We are delighted to hear that the Reviewer feels that the paper has been greatly improved. We thank the reviewer again for their effort in assessing our manuscript and helping us greatly improve our manuscript!

Reviewer #4:

The authors addressed all issues raised during the revision. This is a solid study giving strong contributions to the gene therapy field for cystic fibrosis. I suggest accepting for publication.

We are glad that we have fully addressed the reviewer's comments and concerns and the paper can now be accepted for publication. We thank the Reviewer again for taking time to help us improve our research work!